# Intact *Drosophila* central nervous system cellular quantitation reveals sexual dimorphism

**Wei Jiao[1], Gard Spreemann[1†], Evelyne Ruchti[1], Soumya Banerjee[1], Samuel Vernon[1], Ying Shi[1], R Steven Stowers[2], Kathryn Hess[1], Brian D McCabe[1]***

[1]Brain Mind Institute, EPFL - Swiss Federal Institute of Technology, Lausanne, Switzerland; [2]Department of Microbiology and Cell Biology, Montana State University, Bozeman, United States

**\*For correspondence:**
brian.mccabe@epfl.ch

**Present address:** [†]Telenor Research, Fornebu, Norway

**Competing interest:** The authors declare that no competing interests exist.

**Abstract** Establishing with precision the quantity and identity of the cell types of the brain is a prerequisite for a detailed compendium of gene and protein expression in the central nervous system (CNS). Currently, however, strict quantitation of cell numbers has been achieved only for the nervous system of *Caenorhabditis elegans*. Here, we describe the development of a synergistic pipeline of molecular genetic, imaging, and computational technologies designed to allow high-throughput, precise quantitation with cellular resolution of reporters of gene expression in intact whole tissues with complex cellular constitutions such as the brain. We have deployed the approach to determine with exactitude the number of functional neurons and glia in the entire intact larval *Drosophila* CNS, revealing fewer neurons and more glial cells than previously predicted. We also discover an unexpected divergence between the sexes at this juvenile developmental stage, with the female CNS having significantly more neurons than that of males. Topological analysis of our data establishes that this sexual dimorphism extends to deeper features of CNS organisation. We additionally extended our analysis to quantitate the expression of voltage-gated potassium channel family genes throughout the CNS and uncover substantial differences in abundance. Our methodology enables robust and accurate quantification of the number and positioning of cells within intact organs, facilitating sophisticated analysis of cellular identity, diversity, and gene expression characteristics.

## Editor's evaluation

This manuscript describes a pipeline involving whole brain imaging, automated neuronal segmentation, and topographical analysis, to assess the number of specific cell types in the larval *Drosophila* brain. The authors uncover unexpected sexual dimorphism at this early stage. This paper will be of interest to neuroscientists – from those who use larval *Drosophila* as their study model to others who are generally interested in connectomics and transcriptomics.

## Introduction

Establishing the precise number of cells in the brain is essential to create organ-wide catalogues of cell types and their gene expression (*Lent et al., 2012*; *Devor et al., 2013*). However, apart from the nervous system of the nematode *Caenorhabditis elegans* (302 neurons, 56 glia) (*White et al., 1986*), the exact numbers of cells within the central nervous system (CNS) of model organisms or that of humans is currently unknown, with estimates, including those based on extrapolation from direct

quantification of brain sub-regions, varying widely (*Silbereis et al., 2016*; *Keller et al., 2018*; *von Bartheld et al., 2016*).

Studies of the CNS of *Drosophila melanogaster*, which in scale and behavioural repertoire has been viewed as intermediate between nematodes and rodents (*Bellen et al., 2010*; *Alivisatos et al., 2012*), currently include large-scale efforts to establish both a neuronal connectome and a cell atlas (*Scheffer and Meinertzhagen, 2019*; *Allen et al., 2020*; *Li et al., 2022*). Nonetheless, the precise number of cells (neurons or glia) in either the smaller larval or larger adult *Drosophila* CNS, comprised of both a brain and ventral nerve cord (VNC), remain unknown, though several approximations have been suggested. For the larval CNS, a range of 10,000–15,000 active neurons has been proposed (*Scott et al., 2001*; *Meinertzhagen, 2018*; *Eschbach and Zlatic, 2020*) across developmental time points. For adult *Drosophila*, approximations have been suggested in the range of 100,000–199,000 neurons in the brain (*Simpson, 2009*; *Chiang et al., 2011*; *Kaiser, 2015*; *Scheffer and Meinertzhagen, 2019*; *Raji and Potter, 2021*) together with a range of 10,000–20,000 cells in the VNC (*Birkholz et al., 2015*; *Lacin et al., 2019*; *Bates et al., 2019*; *Allen et al., 2020*). The other major CNS cell type, glia, has been estimated to be approximately 10% of the number of neurons (*Kremer et al., 2017*; *Meinertzhagen, 2018*; *Raji and Potter, 2021*). Given the large diversity of these estimates, precise quantification of the numbers of *Drosophila* neurons and glia would seem a desirable goal, beginning with the smaller larval CNS, which enables the wide compendium of larval *Drosophila* behaviours (*Gerber et al., 2009*; *Neckameyer and Bhatt, 2016*; *Eschbach and Zlatic, 2020*; *Louis, 2020*; *Gowda et al., 2021*).

Complicating the aspiration to quantitate the *Drosophila* larval CNS, in addition to the general problem of separating and quantifying primary cell types such as neurons and glia, are two specific confounding factors that limit simple total cell quantification approaches. First, encompassed within and surrounding the larval CNS are dividing neuroblasts, which will give rise to adult neurons (*Doe, 2017*). Relatedly, imbedded within the larval CNS are substantial numbers of immature adult neurons, observed from electron micrograph reconstructions as having few or no dendrites and axons that terminate in filopodia lacking synapses (*Eichler et al., 2017*). These immature neurons are unlikely to contribute to larval CNS function and are generally excluded when considering larval neuronal circuit architecture (*Eichler et al., 2017*; *Scheffer and Meinertzhagen, 2019*). It has been suggested that only a small fraction of the total number of larval CNS cells may actually contribute to CNS function (*Ravenscroft et al., 2020*).

Here, we have sought to develop a synergistic molecular genetic, imaging, and computational pipeline designed de novo to allow automated neuron, glia, or other gene expression features to be precisely quantitated with cellular resolution in an intact whole CNS. Central to the approach are high signal-to-noise gene expression reporters that produce a punctate, nucleus-localised output, facilitating downstream automated computational measurements and analysis. Exploiting multiple genetic reagents designed to selectively identify only functional neurons with active synaptic protein expression, we identify substantially fewer neurons than most previous estimates in the *Drosophila* larval CNS and, in addition, substantially more glia. We also discover a previously unsuspected sexual dimorphism in the numbers of both cell types at larval stages. The generation of whole CNS point clouds from our data enables us to apply the tools of topological data analysis (TDA) to summarise the CNS in terms of multiscale topological structures. Utilisation of these topological summaries in a support vector machine also supports that sexual dimorphism extends to deeper features of CNS organisation. Finally, we applied our pipeline to quantitate the whole CNS expression frequency of the *Drosophila* family of voltage-gated potassium channels, which revealed divergent channel expression frequencies throughout the CNS. We envision that our method can be employed to allow precise quantitation of gene expression characteristics of the constituent cells of the brain, and potentially other intact whole organs, in a format suitable for sophisticated downstream analysis.

## Results

### Genetic and imaging tools designed to facilitate automated whole CNS cellular quantitation

To establish a robust quantitative method to measure gene expression frequency and quantify the number of cells that contribute to *Drosophila* larval CNS function, we sought to develop a pipeline

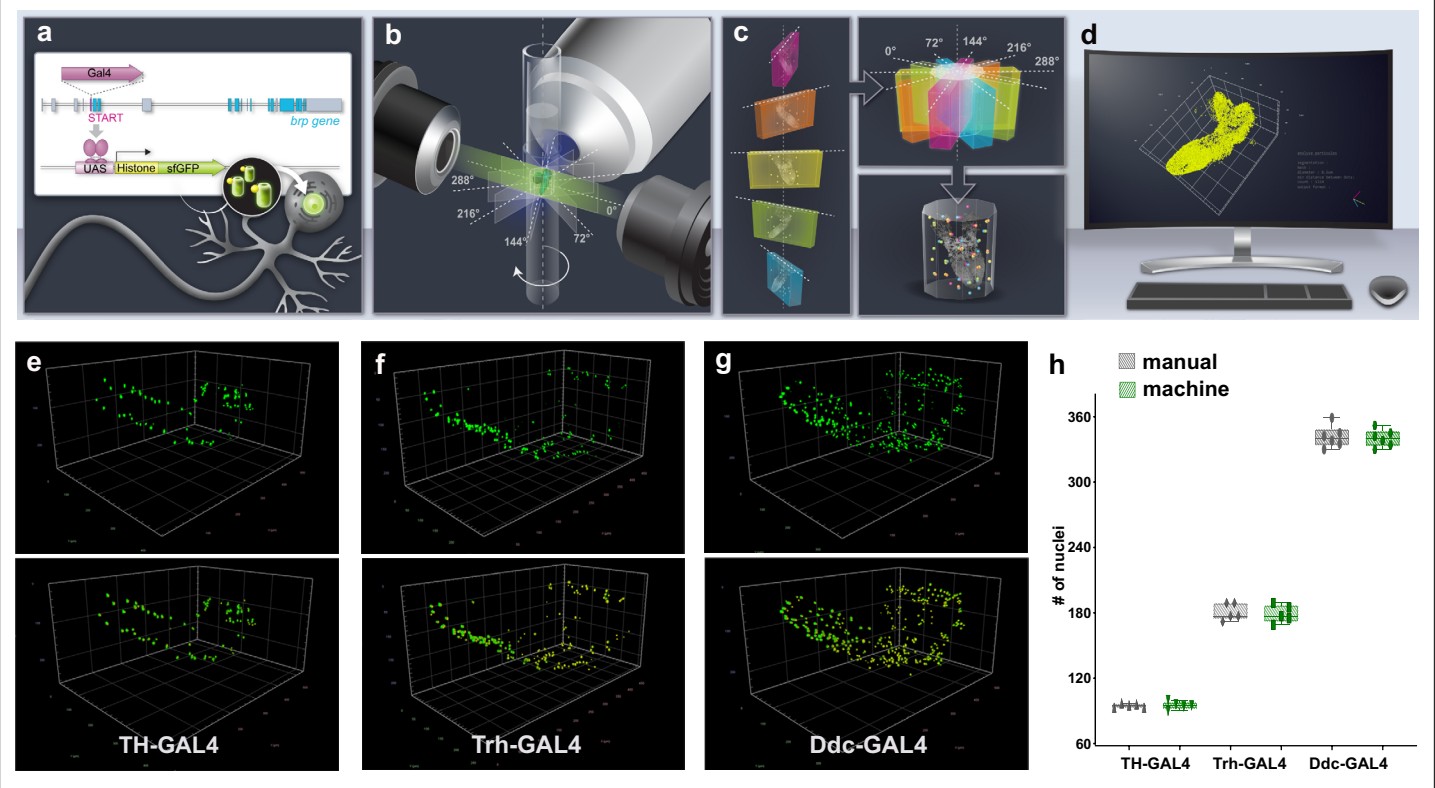

**Figure 1.** Intact whole CNS quantitation pipeline schematic and validation. (**a–d**) Illustration of an intact whole central nervous system (CNS) genetic, imaging and computational pipeline. (**a**) Genetic reagents: GAL4 is introduced into the exons of genes encoding synaptic proteins (e.g. bruchpilot [*brp*]) to capture their expression pattern with high fidelity. GAL4 expression regulates the production of UAS fluorescent-histone reporters, which target to the nucleus of cells, producing a punctate signal. (**b**) Imaging: the intact CNS is imaged at high resolution using light-sheet microscopy. Images are captured at five different angles at 72° intervals. (**c**) Assembly: multiview light sheet images are registered, fused, and deconvolved. (**d**) Quantitation: the volume is segmented, and the nucleus number and relative position are measured. Three-dimensional coordinates of the geometric centre of every nucleus can be calculated to produce a point cloud of nuclei positions.(**e–h**) Pipeline validation. Three-dimensional images before segmentation (above) and subsequent to segmentation (below) of (**e**) dopaminergic (*TH*-GAL4) neurons, (**f**) serotonergic neurons (*Trh*-GAL4) and (**g**) dopa decarboxylase expressing (*Ddc*-GAL4) neurons. (**h**) Manual or automated quantification of nuclei numbers in these volumes are similar. Scale squares in (**e**) and (**g**) are 100 μm and in (**f**) is 50 μm. (**h**) Bars indicate minimum and maximum values.

The online version of this article includes the following source data for figure 1:

**Source data 1.** Source Data for *Figure 1*.

utilising genetic reporters designed to expedite automated neuron and glia quantitation from three-dimensional intact organ images. While membrane-associated reporters are generally employed to label *Drosophila* neurons (*Pfeiffer et al., 2008*; *Jenett et al., 2012*; *Ravenscroft et al., 2020*), we generated UAS-driven (*Brand and Perrimon, 1993*; *Wang et al., 2012*) fluorescent reporters fused to histone proteins (*Sherer et al., 2020*) to target fluorescence only to the nucleus, in order to facilitate subsequent automated segmentation and counting. Through empirical selection of transgene genomic integration sites, we established a set of reporter lines that produced a strong and specific punctate nucleus signal when expression was induced, with little to no unwanted background expression. We then developed a procedure to capture the entire microdissected larval CNS volume by light sheet microscopy at multiple angles and with high resolution, imaging only animals within the ~2 hr developmental time

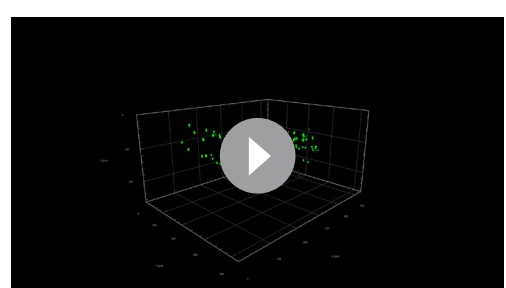

**Video 1.** Larval CNS labelled with *TH*-GAL4.
https://elifesciences.org/articles/74968/figures#video1

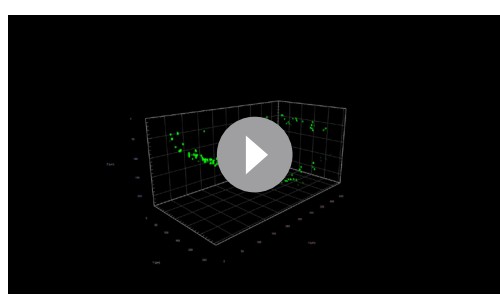

**Video 2.** Larval CNS labelled with *Trh*-GAL4.
https://elifesciences.org/articles/74968/figures#video2

window of the wandering third instar larval stage (*Ainsley et al., 2008*). These multiview datasets were then processed to register, fuse, and deconvolve the entire larval CNS volume. The volume was then segmented and cell numbers were automatically quantified (*Figure 1a–d*).

To evaluate the reliability of the procedure, we began by comparing automated counts of distinct neuronal subtypes with manual counting. We separately labelled all dopaminergic neurons (*Figure 1e*, *TH*-GAL4, *Video 1*; *Friggi-Grelin et al., 2003a*; *Mao and Davis, 2009*), serotonergic neurons (*Figure 1f*, *Trh*-GAL4, *Video 2*; *Alekseyenko et al., 2010*) and neurons that produce both types of neurotransmitter (*Figure 1g*, *Ddc*-GAL4, *Video 3*; *Lundell and Hirsh, 1994*) in the larval CNS. Quantification revealed a high level of concordance (*Figure 1h*, +/-0.21%, n=5 for *TH*-GAL4, +/-1%, n=5 for *Trh*-GAL4, +/-0.38%, n=6 for *Ddc*-GAL4, *Figure 1—source data 1*) between automated and manual measurements of these neuronal subtypes establishing confidence in the procedure.

## Number of neurons and glia in the female larval CNS

Encouraged by our neuronal subset quantitation results, we next sought to generate GAL4 lines for genes likely to be expressed only in active larval neurons with synaptic connections but not by neuroblasts or by immature neurons (*Figure 2—figure supplement 1*). We biased towards generating GAL4 insertions within endogenous genomic loci in order to reproduce endogenous patterns of gene expression with high fidelity.

Bruchpilot (Brp) is a critical presynaptic active zone component widely used to label *Drosophila* synapses, including for large-scale circuit analyses (*Wagh et al., 2006*). We employed CRISPR/Cas9 genome editing to insert GAL4 within exon 2 of the *brp* gene, utilising a T2A self-cleaving peptide sequence (*Diao et al., 2015*) to efficiently release GAL4. While this exonic insertion generated a hypomorphic allele of *brp* (data not shown) when homozygous, the line was employed in heterozygotes to capture Brp protein expression with high fidelity. To complement this line, we used the Trojan/MiMIC technique (*Diao et al., 2015*), to generate a GAL4 insertion in the *Syt1* gene, which encodes Synaptotagmin 1 (*Littleton et al., 1994*), the fast calcium sensor for synaptic neurotransmitter release (*Quiñones-Frías and Littleton, 2021*). Lastly, we examined a transgenic line where an enhancer of *nSyb* (*neuronal Synaptobrevin*) (*Deitcher et al., 1998*), which encodes an essential presynaptic vSNARE (*Südhof and Rothman, 2009*), is used to control GAL4 expression (*Aso et al., 2014*). All three lines were expressed in a similar pattern, labelling a substantial fraction but not all of the total cells in the larval CNS (*Figure 2a–c*, *brp*-GAL4 female *Video 4*, *Syt1*-GAL4 female *Video 5*, *nSyb*-GAL4 female *Video 6*). These lines contrasted with the widely used *elav*-GAL4 (*Lin and Goodman, 1994*), which was expressed in larval neurons, but also apparently in some immature neurons and potentially in some glia as well (*Berger et al., 2007*; *Figure 2—figure supplement 1*). To characterise our lines, we examined their expression throughout development, beginning with embryogenesis. We detected no expression from any of the three lines prior to embryonic stage 16 (*Figure 2—figure supplement 2a*). However, beginning at stage 17 of embryonic development, when synaptic activity begins (*Baines and Bate, 1998*), all three lines displayed expression in both the CNS and peripheral nervous system (*Figure 2—figure supplement 2a*). We also examined if these lines were expressed in neural stem cells during larval stages by co-labelling the larval CNS with the transcription factor Deadpan, a neuroblast marker (*Bier et al., 1992*). We found that labelling by all three

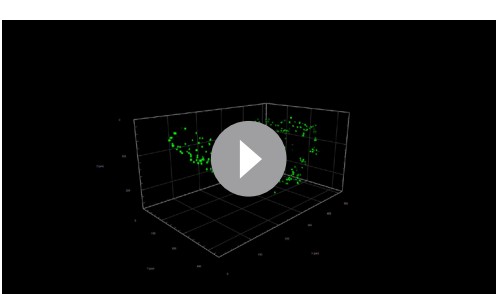

**Video 3.** Larval CNS labelled with *Ddc*-GAL4.
https://elifesciences.org/articles/74968/figures#video3

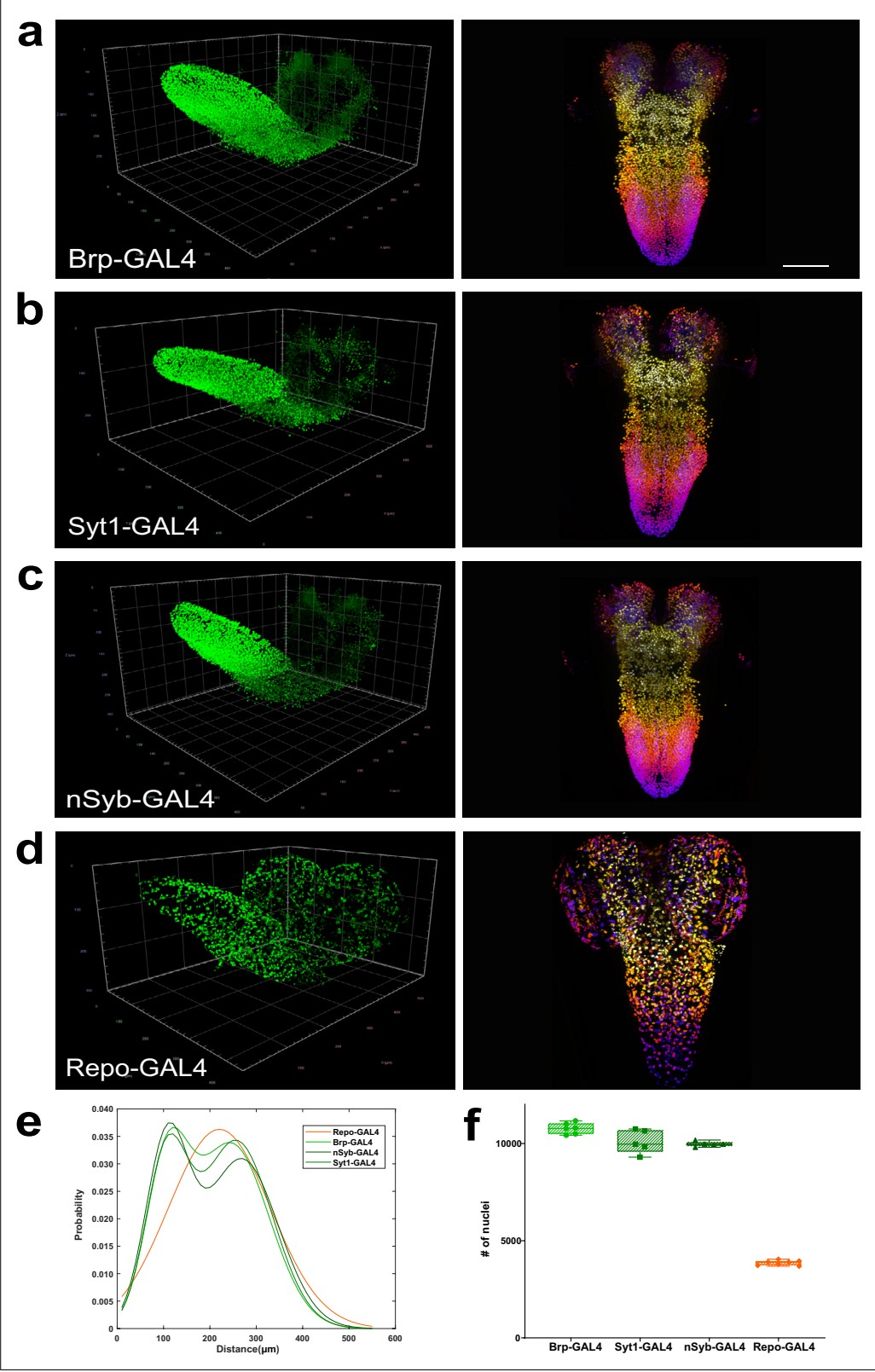

**Figure 2.** Quantitation of neurons and glia in the whole female larval CNS. (**a–d**) Multiview deconvolved images (left) and z-stack projections (right) (colours represent z position) of the central nervous system (CNS) of (**a**) *brp*-GAL4, (**b**) *Syt1*-GAL4, (**c**)*nSyb*-GAL4, and (**d**) *repo*-GAL4. (**e**) Distribution of inter-nuclei distances for each line. (**f**) Quantification of the number of labelled nuclei in each line. (**a–d**) left; scale squares (**a**) and (**c**) = 50 µm, (**b**) and

*Figure 2 continued on next page*

*Figure 2 continued*

(**d**) = 100 μm; right images identical magnification, scale bar = 100 μm. (**f**) Bars indicate minimum and maximum values.

The online version of this article includes the following source data and figure supplement(s) for figure 2:

**Source data 1.** Source Data for *Figure 2*.

**Figure supplement 1.** Larval CNS stem cells.

**Figure supplement 2.** Developmental expression of neuronal GAL4 lines.

lines did not overlap with Deadpan expression (*Figure 2—figure supplement 2b*), suggesting these lines are not expressed in neuroblasts. We also examined expression of all three lines in the adult brain and, as in the larval CNS, observed labelling of a large fraction but not all of the total cells in the adult brain (*Figure 2—figure supplement 2c*). Lastly, to ensure that the cells labelled by our lines were exclusively neurons, we compared their expression to that of glial cells labelled by glial specific transcription factor Repo (*Xiong et al., 1994*; *Lin and Potter, 2016*) using independent and mutually exclusive QF2 dependent labelling. We found complete exclusion of cells labelled by *brp*, *Syt1* and *nSyb* GAL4 lines from cells labelled by *repo* (*Figure 2d*, *brp*-GAL4 & *repo*-QF2 *Video 7*, *Syt1*-GAL4 & *repo*-QF2 *Video 8*, *nSyb*-GAL4 & *repo*-QF2 *Video 9*), consistent with the *brp*, *Syt1* and *nSyb* GAL4 lines labelling only neurons that express synaptic protein genes and not glial cells.

To further compare these lines, beginning with the CNS of female animals, we calculated three-dimensional coordinates for the geometric centre of all nuclei labelled in the *brp*, *Syt1* and *nSyb* GAL4 lines to generate point cloud mathematical objects and compared them to point clouds of glial nuclei labelled by the *repo*-GAL4 line. We then plotted and compared the distributions of inter-nuclei distances in these lines. Using this measurement, we found that the inter-nuclei distance of glial cell nuclei exhibited a unimodal distribution (*Figure 2e*). In contrast, all three neuronal lines exhibited a bimodal distribution of inter-nuclei distances (*Figure 2e*). We thus observed two patterns of labelled nuclei, one shared among neuronal lines and the other distinct for glia (*Figure 2e*), again consistent with these lines labelling different cell types.

We next counted the number of nuclei labelled by these neuronal and glial lines, again beginning with females (*Figure 2f*). We found that the CNS labelled by *brp*-GAL4 had 10,776 (±2.65%, n=6) neurons, *Syt1*-GAL4 had 10,097 (±5.96%, n=5) neurons, and *nSyb*-GAL4 had 9971 (±1.35%, n=5) neurons (*Figure 2f*). We tested the statistical difference in the numbers of neurons labelled by these lines and found that while *nSyb*-GAL4 and *Syt1*-GAL4 were not statistically different from each other, *brp*-GAL4 did label significantly more neurons than either *Syt1* or *nSyb* GAL4 lines (*brp*-GAL4 vs *Syt1*-GAL4+6.72%, p=0.03, *brp*-GAL4 vs *nSyb*-GAL4 +8.07%, p=0.01). Averaging across the lines, we found that the female third instar larval CNS had 10,312 ±5.03%, n=16, neurons (*Figure 2— source data 1*). To ensure that our method did not introduce bias in dense datasets, we also manually counted a *brp*-GAL4 labelled CNS and compared it to the automated count. Similar to our experiments with sparse neuronal labelling, we found good agreement between manual and automated quantification with a difference of just 14 neurons (9430 nuclei manual vs 9444 nuclei automated for this individual CNS).

We next counted the number of glia labelled by the *repo*-GAL4 line (*Figure 2d and f*, *repo*-GAL4 female, *Video 10*). We measured 3860 ±3.37%, n=7, glia in the female CNS (*Figure 2—source data 1*).

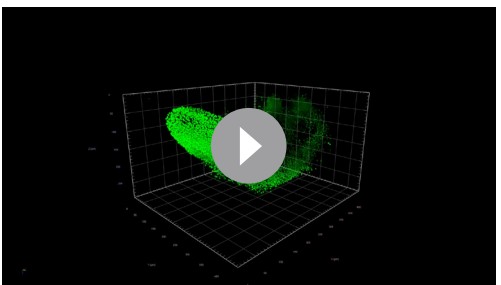

**Video 4.** Female CNS labelled with *brp*-GAL4.
https://elifesciences.org/articles/74968/figures#video4

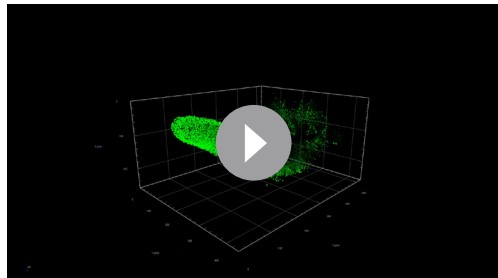

**Video 5.** Female CNS labelled with *Syt1*-GAL4.
https://elifesciences.org/articles/74968/figures#video5

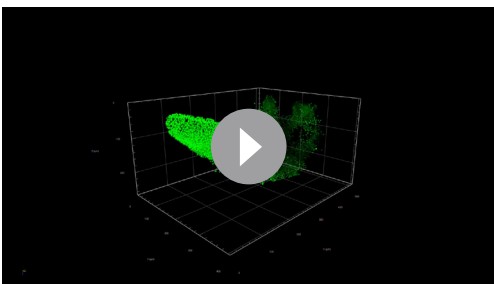

**Video 6.** Female CNS labelled with *nSyb*-GAL4.
https://elifesciences.org/articles/74968/figures#video6

This amounted to 37% of the number of neurons, far more than previously estimated (**Meinertzhagen, 2018**; **Raji and Potter, 2021**). In sum, we found that the female *Drosophila* larval CNS had 10,312 neurons, fewer than most previous predictions, and several fold more glia than previously thought.

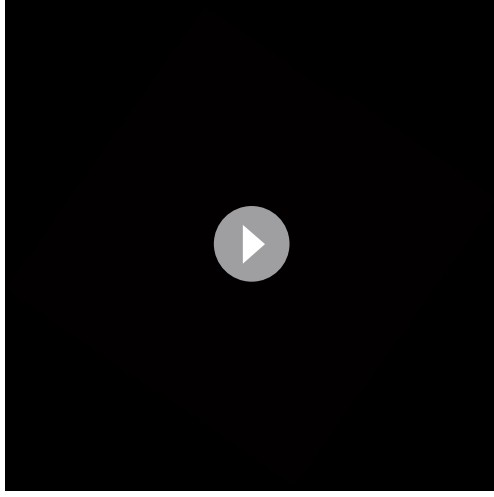

**Video 8.** Larval CNS labelled with *Syt1*-GAL4 and *repo*-QF2.
https://elifesciences.org/articles/74968/figures#video8

## Males have fewer neurons and more glia than females

We next carried out a similar analysis on the CNS of male larvae (*Figure 3a–c*). We found that *brp*-GAL4 labelled 9888 (±3.15%, n=5) neurons, *Syt1*-GAL4 labelled 9012 (±3.8%, n=5) neurons, and *nSyb*-GAL4 labelled 9286 (±5.38%, n=5) neurons in male larvae (*Figure 3e*, *Figure 3—source data 1*). In males, *brp*-GAL4 did not label significantly more neurons than *nSyb*-GAL4 but did label more than *Syt1*-GAL4 (*brp*-GAL4 vs *Syt1*-GAL4 +9.72%, p=0.01), while the number of neurons labelled by *nSyb*-GAL4 was not significantly different from *Syt1*-GAL4, similar to what we had found in females. Averaging across the lines, we found that the male third instar larval CNS had 9396 ±5.59%, n=15 neurons, significantly fewer than those of females (–9.75%, p<0.0001). This difference was also consistent within individual genotypes with *brp*-GAL4 labelling (–8.98%, p=0.0008), *Syt1*-GAL4 labelling (–12.04%, p=0.008) and *nSyb*-GAL4 labelling (–7.38%, p=0.0182) less neurons in males than in females.

We also counted the number of glia labelled by *repo*-GAL4 in males (*Figure 3d and e*). We found that males had 4015 ±1.98%, n=6, glia far more than previous estimates (*Figure 3—source data 1*). The number of glia in the male larval CNS was significantly more than in females (+3.86%, p=0.0284).

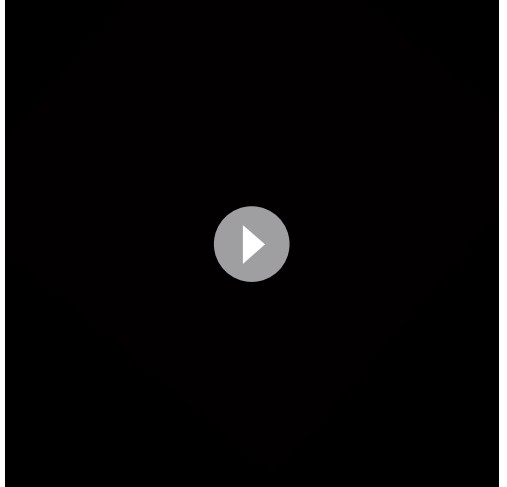

**Video 7.** Larval CNS labelled with *brp*-GAL4 and *repo*-QF2.
https://elifesciences.org/articles/74968/figures#video7

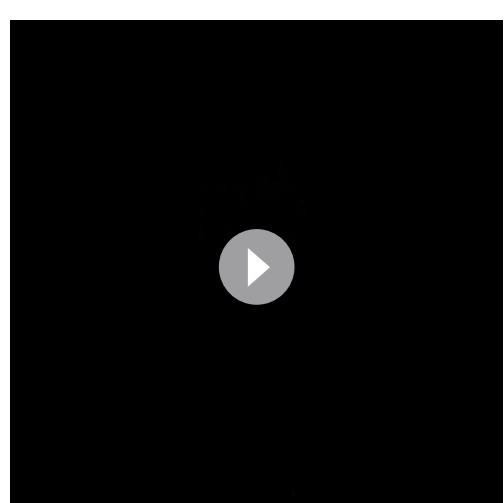

**Video 9.** Larval CNS labelled with *nSyb*-GAL4 and *repo*-QF2.
https://elifesciences.org/articles/74968/figures#video9

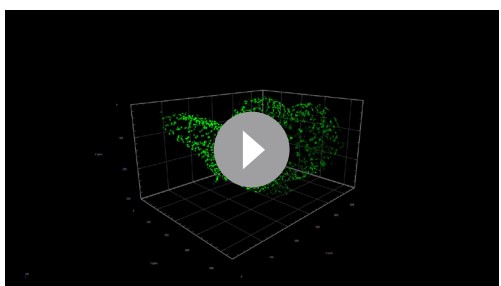

**Video 10.** Female CNS labelled with *repo*-GAL4.
https://elifesciences.org/articles/74968/figures#video10

In summary, male *Drosophila* larva have significantly fewer CNS neurons than females but more glia.

## Topological analysis detects CNS structural differences between males and females

We next wished to determine if the differences between point clouds derived from the positions of neuronal nuclei of the male and female CNS went beyond simple numerics. To do this, we applied the tools of TDA (*Rabadán and Blumberg, 2019*; *Chazal and Michel, 2021*) to summarise the CNS in terms of multiscale topological structures (*Expert et al., 2019*). These topological summaries, the construction of which is described in the methods, can be thought of as multiscale descriptions of the shape of the dataset. Topological summaries, which can be compared by standard methods despite the lack of common reference points, could then be used as the classification features in a support vector machine (SVM). Since the total number of point clouds was relatively small for this type of analysis (*Supplementary file 1*), we down-sampled each whole CNS point cloud randomly to 8000 points 100 times, producing a total of 3100 point clouds, for each of which we then computed a certain topological summary, called the *degree-1 persistence diagram of its alpha complex* (*Edelsbrunner and Mücke, 1994*).

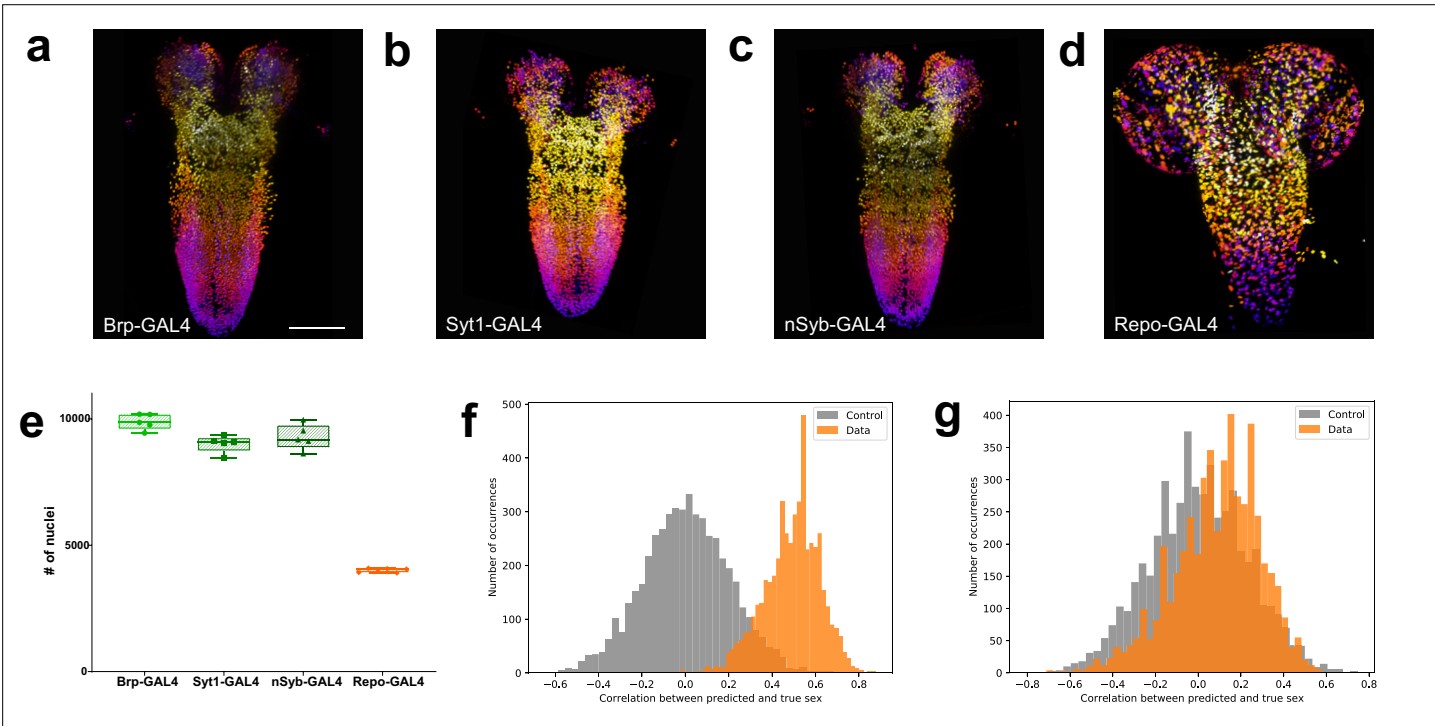

**Figure 3.** Quantitation of neurons and glia in the male larval CNS and topological comparison of sex differences. (**a–d**) Example z-stack projections (colours represent z position) of male larval central nervous system (CNS) of (**a**) *brp*-GAL4, (**b**) *Syt1*-GAL4, (**c**) *nSyb*-GAL4, and (**d**) *repo*-GAL4. (**e**) Quantification of the number of labelled nuclei in each line. (**f**) The distribution of correlations between the ground truth and the prediction made by the support vector machine (SVM) using topological features is indicative of sexual dimorphism of the higher order structure of neuron point clouds (**g**) Simpler point cloud features such as properties of the distributions of inter-nuclei distances are not indicative of this. (**a–d**): identical magnification, scale bar = 100 µm. (**e**) Bars indicate minimum and maximum values.

The online version of this article includes the following source data for figure 3:

**Source data 1.** Source Data for *Figure 3*.

After fixing the necessary hyperparameters, sex classification experiments were run across 5000 random train/test splits of the topological summaries. In each split, the summaries derived from subsamplings of a single CNS point cloud were either all in the training set or all in the testing set, to avoid leaking information. Each time, the SVM was trained once with the animal's true sex as the target class and once with a randomly assigned sex as the target as a control. We then computed the Pearson's correlation between the classifier's output on the testing set and the true (respectively randomised) sex of the animal.

The 5000 splits were used to produce 5000 correlations with the true sex and 5000 correlations with a randomly assigned sex. The distribution of these correlations (*Figure 3f*), exhibiting clearly that the SVM is able to extract the sex of the animal reliably: only about 1.9% of the splits result in a higher correlation in the control set than in the true data. Moreover, repeating the procedure with simpler point cloud features, like properties of the distributions of inter-nuclei distances, did not produce a significant signal (*Figure 3g*). Thus, the pattern, which seems hard to describe concisely, is not revealed through simpler descriptors of the neuron configurations, leading us to suspect that CNS sexual dimorphism extends to deeper features of organisation that are both subtle and widely distributed. These results, in addition to the differences in total cell numbers, support sexual dimorphism of the male and female *Drosophila* CNS at the larval stage.

## Potassium channel family member gene expression density in the CNS

Having established a baseline of total numbers of neurons in the larval CNS, we next sought to deploy the quantification pipeline to measure the expression frequency of key neuronal function genes throughout the CNS. We chose to examine the family of voltage-gated potassium channels, which are essential for many aspects of neuronal function and for which *Drosophila* studies defined the founding members (*McCormack, 2003*). We generated GAL4 insertions in the Shaker (*Sh*) (Kv1 family), Shab (*Shab*) (Kv 2 family), Shaw (*Shaw*) (Kv3 family), and Shal (*Shal*) (Kv4 family) (*McCormack, 2003*) genes using the Trojan/Mimic technique (*Diao et al., 2015*). As the *Sh* gene is x-linked, we carried out our quantitation analysis in the male CNS only to avoid potential gene dosage effects. To determine whether our GAL4 reporter lines had patterns of expression consistent with the known properties of these channels, we examined the expression of all four lines in motor neurons, where functional activity for Shaker, Shab, Shaw, and Shal had previously been demonstrated by electrophysiological measurements (*Covarrubias et al., 1991*; *Ryglewski and Duch, 2009*). We found that the GAL4 reporters for all 4 channels were expressed as expected in motor neurons (*Figure 4—figure supplement 1*), consistent with accurate reproduction of the established expression of these proteins.

We next examined the expression frequency of these genes in the entire CNS (*Figure 4a–d*, *Sh*-GAL4 *Video 11*, *Shal*-GAL4 *Video 12*, *Shab*-GAL4 *Video 13*, *Shaw*-GAL4 *Video 14*). We found that *Sh* and *Shal* were expressed in large numbers of neurons 8204 ±5.67%, n=10 and 8261 ±3.1%, n=5 respectively, though significantly less (−12.7% and −12.1%, p<0.0001) than the average number of all male neurons (*Figure 4a, b and e*, *Figure 4—source data 1*). In contrast, *Shab* (3057 ±8.21%, n=10) and *Shaw* (1737 ±4.3%, n=11) were expressed in smaller numbers of neurons (*Figure 4c–e*), with expression observed in only 32.5% or 18.5% of total male neurons respectively, suggesting more discrete functions within CNS neurons, contrasting with the collective expression of all four genes within motor neurons (*Figure 4—figure supplement 1*). In particular, *Shab* and *Shaw* had very reduced expression in the brain lobes of larva (*Figure 4c and d*) compared with *Sh* and *Shal* (*Figure 4a and b*). These results establish that our genetic-imaging pipeline can enable quantitation of the expression frequency of families of genes essential for neuronal properties across the entire CNS.

## Discussion

Establishing the number and identity of cells in the CNS is a foundational metric upon which to construct molecular, developmental, connectomic, and evolutionary atlases of central nervous systems across species (*Lent et al., 2012*; *Devor et al., 2013*). Here, we develop and deploy a methodological pipeline to label discrete cell types in the intact *Drosophila* CNS with genetic reporters designed to facilitate the subsequent segmentation and automated quantification of cell types, in addition to capturing positional coordinates of relative nucleus positions throughout the organ. Using this toolset, we find fewer active neurons, as defined by expression of synaptic protein genes, in the *Drosophila* larval CNS

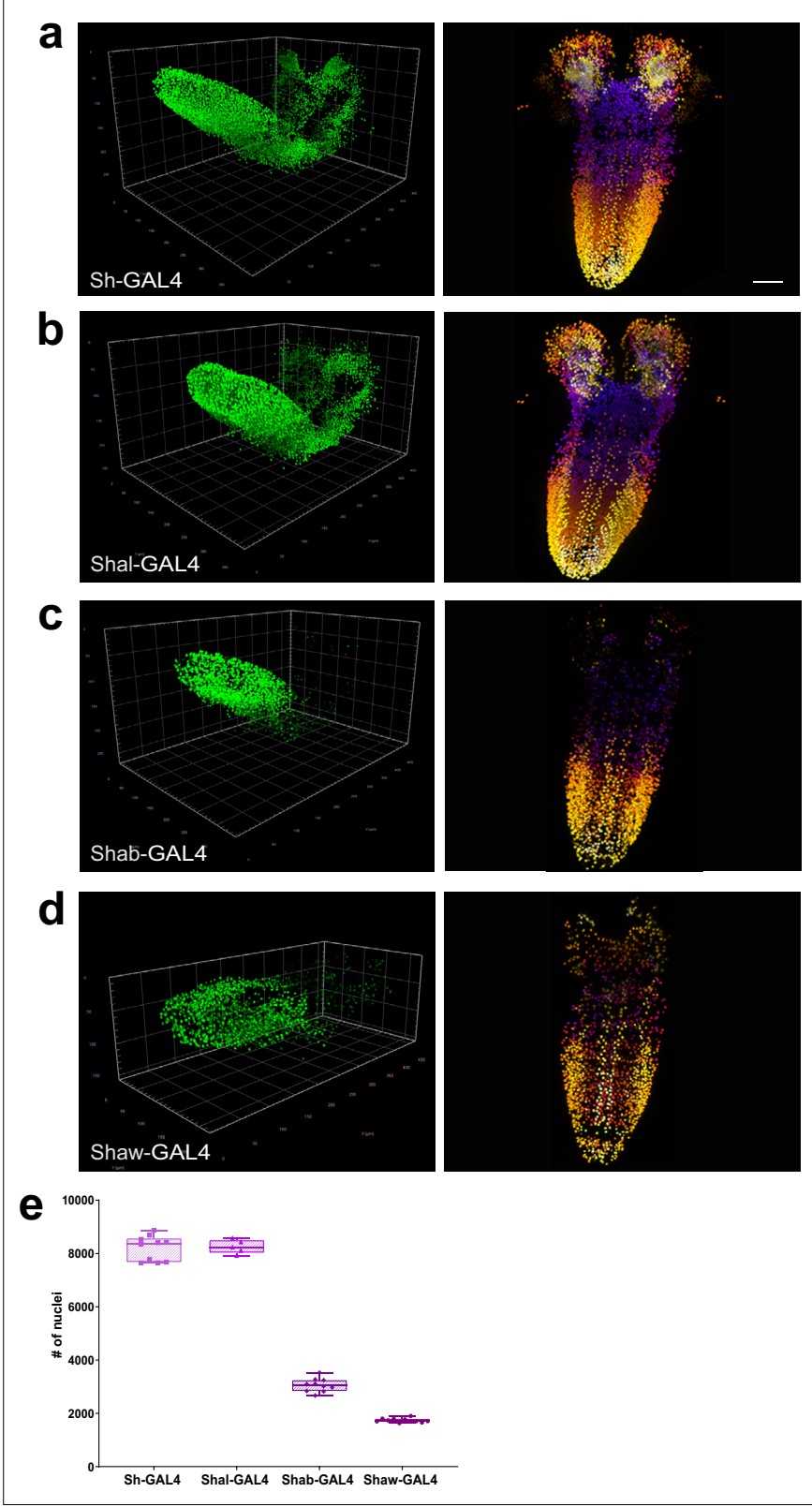

**Figure 4.** Quantitation of the number of neurons expressing voltage-gated potassium channel genes. (**a–d**) Multiview deconvolved images (left) and z-stack projections (right) (colours represent z position) of potassium channel family members: (**a**) *Sh*-GAL4, (**b**) *Shal*-GAL4, (**c**) *Shab*-GAL4, and (**d**) *Shaw*-GAL4. (**e**) Quantification of the number of labelled nuclei in each line. (**a–d**) left, scale squares = 50 µm, right, identical magnification, scale bar =

*Figure 4 continued on next page*

*Figure 4 continued*

50 µm. (**e**) Bars indicate minimum and maximum values.

The online version of this article includes the following source data and figure supplement(s) for figure 4:

**Source data 1.** Source Data for *Figure 4*.

**Figure supplement 1.** Expression of voltage gated K+ channel GAL4 lines in motor neurons.

than most previous predictions and also substantially more glia. We additionally discover previously unsuspected differences in both neuron and glial density and CNS topology between the sexes at the larval stage, when external sex organs are absent, with females possessing both more neurons and fewer glia than males. Topological analysis of point clouds derived from neuronal nucleus position, which can detect potentially subtle and complex geometric structure in the data, also strongly support the existence of differences between the male and female CNS. In addition, deploying these tools, we find that while all members of the *Drosophila* voltage-gated potassium channel family are expressed in motor neurons, consistent with prior mutant analyses, the Kv2 channel Shab and Kv3 channel Shaw are expressed in a much smaller number of neurons in the CNS than the Kv1 channel Shaker and the Kv4 channel Shal, suggesting conclusions drawn about the coordinated activity of these channels from studies of motor neurons may not be broadly applicable across the CNS, where the genes encoding these channels are frequently not co-expressed.

A number of semiquantitative methods have been employed to estimate the number of neurons in the brains of humans and model organisms, including *Drosophila* (*Lent et al., 2012*; *Keller et al., 2018*). For example, the number of neurons or other cells in the brain has been estimated using stereological counting of subregions. A major limitation of this approach is the assumption of homogenous cell density across the organ or within subregions, which is not supported by the high variability of counts even between samples of similar regions, and thus likely introduces large errors (*von Bartheld et al., 2016*; *Keller et al., 2018*). Rough extrapolation of neuronal counts of electron microscope volumes of regions of the *Drosophila* larval CNS had suggested an estimate of ~15,000 neurons (*Meinertzhagen, 2018*; *Eschbach and Zlatic, 2020*). An alternate approach is isotropic fractionation, where all cells in large regions or the entire CNS are dissociated to produce a homogeneous single-cell suspension. Nuclei in the suspension can then be labelled by immunohistochemistry and cells in a subvolume counted in a Neubauer chamber to estimate the total number of cells present. Limitations of the approach include the necessity to ensure complete dissociation of cells while avoiding tissue loss, the requirement for homogenous antibody labelling, and highly accurate dilution (*Deniz et al., 2018*). This approach has recently been used to estimate the total number of neurons and glia in the adult *Drosophila* brain and suggested a number of 199,000 neurons (*Raji and Potter, 2021*), twice prior estimates (*Scheffer and Meinertzhagen, 2019*; *Allen et al., 2020*). In contrast to our results in the larval CNS, this study found no significant differences in the number of neurons between the sexes and also found that 'non-neuronal' cells, which should include glia, accounted for less than 9% of the total cells counted. In addition to the inherent inaccuracy of the isotropic fractionation technique, which the authors both observed and acknowledge (*Raji and Potter, 2021*), their use of anti-Elav antibody labelling, which can label some glia in addition to neurons (*Berger et al., 2007*), or perhaps differences in life stage, may explain some of the discrepancies between our results.

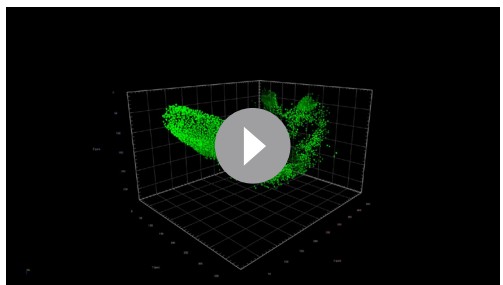

**Video 11.** Larval CNS labelled with *Sh*-GAL4.
https://elifesciences.org/articles/74968/figures#video11

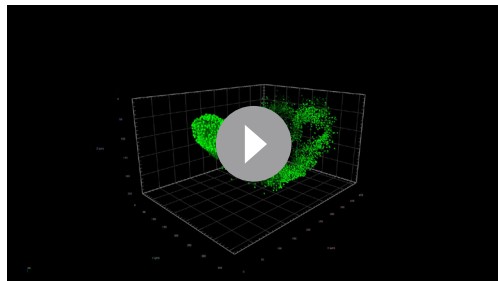

**Video 12.** Larval CNS labelled with *Shal*-GAL4.
https://elifesciences.org/articles/74968/figures#video12

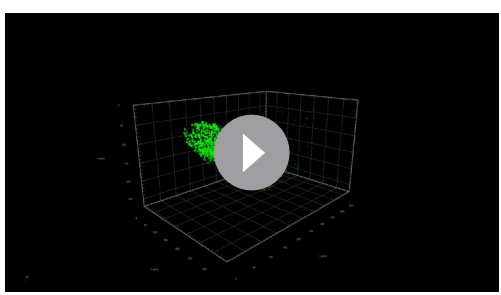

**Video 13.** Larval CNS labelled with *Shab*-GAL4.
https://elifesciences.org/articles/74968/figures#video13

An unpredicted result from our whole CNS neuron quantitation was substantial differences in neuron and glial numbers between the sexes in larva. In adult *Drosophila*, sexually dimorphic neural circuitry has been observed in the olfactory system (*Kimura et al., 2005*), and human females have also been reported to have more olfactory bulb neurons and glia than males (*Oliveira-Pinto et al., 2014*). While sex-specific behavioural differences are obvious in adult *Drosophila* (*Jazin and Cahill, 2010*), few sexually dimorphic behavioural differences have been reported in larva (*Aleman-Meza et al., 2015*). However, male and female larva do differ in nutritional preference (*Rodrigues et al., 2015*; *Davies et al., 2018*), which could potentially account for some aspects of the dimorphism we observe. In addition to differences in total cell numbers, our topological methods, which take into account multiscale structure, suggest that differences in CNS structure between the sexes are both subtle (in the mathematical sense) and non-localised in nature, and indeed are not observable with simpler methods of analysis of CNS organisation.

In addition to enabling precise counting of genetically labelled cells, our method allows the relative measurement of discrete cell types or gene expression frequencies throughout the CNS. For example, the relative frequency of glial cells to neurons in the human brain has been long debated (*von Bartheld et al., 2016*) and in the adult *Drosophila* brain it has been suggested there are 0.1 glial per neuron (*Kremer et al., 2017*; *Scheffer and Meinertzhagen, 2019*; *Raji and Potter, 2021*). In the larval *Drosophila* CNS, we found closer to 0.4 glial cells per neuron on average, more similar to the glial-neuron ratios reported for rodents or rabbits (*Verkhratsky and Butt, 2018*). An important potential caveat, however, is that the large relative ratio of glia we observe in the third instar larva could conceivably be glia produced in advance of adult CNS development. As adult specific neuron numbers expand during pupation, the relative ratio of glia could potentially decline. Additional glial-neuron ratio measurements in the adult CNS will be required to examine this possibility.

Our approach may also allow the assignment of potential functional classes of neurons. For example, from our examination of voltage-gated potassium channel family gene expression, all these channels are collectively expressed in motor neurons; however, the *Shab* and *Shaw* genes have more discrete expression patterns in other CNS neuron classes, potentially imbuing these neurons with unique functional characteristics (*Chow and Leung, 2020*). Future multiplexing of binary genetic expression systems and reporters (*Simpson, 2009*; *del Valle Rodríguez et al., 2011*; *Diao et al., 2015*) should enable neurons or glia to be further quantitively subclassified by gene expression features throughout the entire intact CNS.

## Materials and methods
### *Drosophila* stocks

The following stocks were employed - y[1] w[*]; Mi{y[+mDint2]=MIC}Syt1[MI02197] (BDSC#35973) (*Venken et al., 2011*), y(1) w(*) Mi(y[+mDint2]=MIC) Sh(MI10885) (BDSC#56260), y(1) w(*);Mi(y[+mDint2]=MIC)Shal(MI10881) (BDSC#56089) (*Venken et al., 2011*), y(1) w(*); Mi(y[+mDint2]=MIC) Shab(MI00848) (BDSC#34115) (*Venken et al., 2011*), *nSyb*-GAL4(GMR57C10)(BDSC#39171) (*Pfeiffer et al., 2008*), *repo*-GAL4 (BDSC#7415) (*Sepp et al., 2001*), *repo*-QF2 (BDSC#66477) (*Lin and Potter, 2016*), *Shaw*-GAL4 (BDSC#60325) (*Venken et al., 2011*; *Li-Kroeger et al., 2018*), *Ddc*-GAL4(BDSC#7009) (*Feany and Bender, 2000*),

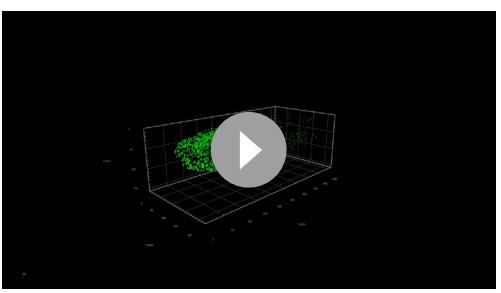

**Video 14.** Larval CNS labelled with *Shaw*-GAL4.
https://elifesciences.org/articles/74968/figures#video14

*TH*-GAL4(BDSC#8848) (*Friggi-Grelin et al., 2003b*), *Trh*-GAL4(BDSC#38389) (*Alekseyenko et al., 2010*), UAS_H2A-GFP (*Sherer et al., 2020*), QUAS_H2B-mCherry (*Sherer et al., 2020*), *brp*-GAL4 (this manuscript), UAS_H2A::GFP-T2A-mKok::Caax (this manuscript). All lines were raised on standard media at 25°C, 50% RH.

## Generation of *brp*-GAL4 exon 2 insertion line

A GAL4.2 sequence was inserted in genome, immediately after the start codon of the Brp-RD isoform using CRISPR based gene editing employing the following constructs. *brp gRNA pCDF3:* two gRNA sequences targeting each side of the insertion location in exon 2 of *brp*, were selected using the FlyCRISPR algorithm (http://flycrispr.molbio.wisc.edu/), consisting of 20 nucleotides each (PAM excluded), and predicted to have minimal off-targets. Each individual 20-nucleotide gRNA sequence were inserted into pCFD3 plasmid (Addgene #49,410) using the KLD enzyme mix (New England Biolabs). *brp-GAL4 insertion construct*: the seven following PCR amplified fragments were assembled using HIFI technology: (1) 1198 bp homology arm covering 5' UTR until 5' target site; (2) the region between 5' target site and the start codon were amplified from *Drosophila* nos-cas9 (attp2) genomic DNA (a modified Pam sequence was inserted using overlapping primers); (3) Linker-T2A-GAL4.2 sequence was amplified from pBID-DSCP-G-GAL4 (*Wang et al., 2012*) (the linker-T2A sequence was added upstream of the forward primer); (4) P10-3'UTR was amplified from pJFRC81-10XUAS-IVS-Syn21-GFP-p10 (Addgene 36432); (5) 3xP3-Hsp70pro-dsRed2-SV40polyA selection cassette, flanked by two LoxP sites, was amplified from pHD-sfGFP scareless dsRed (Addgene 80811); (6) The region covering the end of DsRed cassette until 3' target site; and (7) the 1079 bp homology arm two covering from the 3' target site to exon 2, were amplified from *Drosophila* nos-cas9 (attp2) genomic DNA. Full length assembly was topo cloned in zero-blunt end pCR4 vector (Invitrogen), all constructs have been verified by sequencing (Microsynth AG, Switzerland) and injections were carried out into a nos-cas9 (attp2) strain (*Ren et al., 2013*). Correct insertion of GAL4 was verified by genome sequencing. All primer sequences are included in Appendix 1—key resources table.

## Construction of UAS_H2A::GFP-T2A-mKok::Caax

PCR amplifications were performed using Platinium Superfi polymerase (Invitrogen). The three PCR fragments were assembled together using Hifi technology (Invitrogen): (1) Histone2A (H2A) cDNA was amplified from *pDESTP10 LexO-H2A-GFP* template (Gift from Steve Stowers) with a synthetic 5'UTR sequence (syn21) added upstream to H2A on the forward primer; (2) sfGFP was amplified from template pHD-sfGFP Scareless dsRed (Addgene 80811); and (3) mKok amplified from pCS2 +ChMermaid S188 (Addgene 53617) with the CAAX membrane tag sequence (*Sutcliffe et al., 2017*) added at the 3' end of the protein using the reverse primer. A *Thosea asigna* virus 2 A(T2A) self-cleaving peptide sequence (*Diao et al., 2015*), was inserted between sfGFP and mKok, using sfGFP reverse and mKok forward overlapping primers. The full length assembly was TOPO cloned into pCR8GW-TOPO vector (Invitrogen) generating pCR8GW-H2A::GFP-T2A-mKok::Caax. The insert, H2A::GFP-T2A-mKok::Caax, was, then, transferred to pBID_UASC_G destination vector (*Wang et al., 2012*) using LR II clonase kit (Invitrogen) to generate pBID_UAS-H2A::GFP-T2A-mKok::Caax. The transgene was generated by injection into the JK66B landing site. All primer sequences are included in Appendix 1—key resources table.

## Generation of novel Trojan GAL4 lines

MiMIC lines generated by the group of Hugo Bellen (*Venken et al., 2011*) were acquired from the Bloomington Stock Center. Conversion of Mimic lines to Trojan GAL4 lines was performed as described previously (*Diao et al., 2015*).

## Larval CNS preparation and image acquisition

Wandering third instar larvae were dissected in 1 × PBS (Mediatech) and fixed with 4% formaldehyde (Sigma-Aldrich) for 20 mins. 1 × PBS were added to remove the fixative, and then the CNS was dissected (*Hafer and Schedl, 2006*) and rinsed with 1 × PBS with 4% Triton-X 100 for 2 days at 4°C. After rinses, the CNS was embedded in 1% low melting temperature agarose (Peq gold) mixed with 200 nm red fluorescent beads (1:50,000), then introduced into a glass capillary and positioned well separated from each other. After solidification of the agarose, the capillary was mounted to sample

holder, transferred to a Zeiss Lightsheet Z.1 microscope and the samples were extruded from the capillary for imaging. CNS images were acquired with a 20 ×/1.0 Apochromat immersion detection objective and two 10 ×/0.2 illumination objectives at five different views, with 1 µm z-intervals. Voxel resolution was 0.317 um.

## Image processing and data analysis

Collected multiview datasets were registered and fused with the Fiji Multiview Reconstruction plugin (*Preibisch et al., 2010*; *Schindelin et al., 2012*). Image datasets after multiview deconvolution were analysed with Vision4D 3.0.0 (Arivis AG). A curvature flow filter was first used to denoise the image dataset. Subsequently, a Blob Finder algorithm (*Najman and Couprie, 2003*) was applied to detect and segment bright rounded three-dimensional sphere-like structures in the images with 4.5 µm set as the diameter. Segmented objects with volume less than 15 µm$^3$ were removed from analysis by segmentation filter to avoid unspecific signals. Subsequently, the number of nuclei and the x, y, z coordinates of the geometric center of each nucleus were output from Vision4D. Where manual counting was employed (*Figure 1* and a randomly selected *brp*-GAL4 labelled CNS), Vision4D was used to visualise and iteratively proceed through and manually annotate the dataset. Example whole CNS datasets where functional neurons or glia are labelled are available (*Jiao and McCabe, 2021a*; *Jiao and McCabe, 2021b*). Raw coordinates of the centre of geometry for the nuclei for whole male and female CNS are available in *Supplementary file 1*. In two-dimensional representations, Z position is indicated by colour coding using the scheme below.

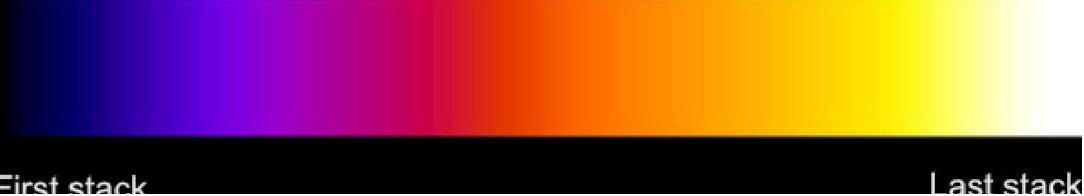

First stack                                                                 Last stack

**Scheme 1.** Z-position colour code employed in 2D representations.

## Mathematical analysis

The topological summaries methods employed have previously been introduced (*Edelsbrunner and Harer, 2010*; *Ghrist, 2014*; *Rabadán and Blumberg, 2019*; *Chazal and Michel, 2021*). For a motivating example of the principles underlying topological summaries, one could think of pearls forming a necklace. Topological summaries express the global structure of the necklace formed by the relationships between the positions of the individual pearls, but are invariant under translation and rotation of the necklace overall. Two such necklaces have topological summaries that are comparable even if the pearls in one have no relationship to the pearls in the other. It is the global structures—such as its circular shape on a large scale, or bulges on a smaller scale—formed by the relationships of the individual pearls of each one that matter.

We trained a machine learning classifier, specifically an SVM, on the CNS nuclei positions in order to evaluate its power in determining characteristics of the animal from which it was derived. The data encompassed all point clouds generated from all CNS lines (*brp*-GAL4, *Syt1*-GAL4, *nSyb*-GAL4, *repo*-GAL4, *Supplementary file 1*). Correlation significance (classification power) is determined by comparing the performance of the SVM on the actual classification task to one where each larva is randomly assigned a class.

Mathematically speaking, the nuclei positions from a single CNS form a point cloud, a finite set of points in R3. A possible, naive approach to SVM feature selection for point clouds would be to consider the mean, variance, or other modes of the distribution of pairwise distances within the cloud. These real-valued features could then be passed through, for example, radial basis function kernels for use in SVMs. We focused on very different kind of features, namely ones obtained from the topology of the point clouds. When the point cloud is of low dimension, such as the three-dimensional point clouds arising from nuclei position data, the following approach is relevant. Let X be a finite point

cloud in R3. For any $r \geq 0$, we let Xr denote the same point cloud, but with each point replaced by a ball of radius r. As r increases, the sequence formed by the Xr expresses different topological features of X. By topological features, we here mean the presence or absence of multiple connected components, unfilled loops, and unfilled cavities.

The figure below illustrates this process in the case of a synthetic two-dimensional point cloud, but the idea extends to any dimension including whole CNS point clouds. When r is small, Xr is topologically very similar to X=X0, and is essentially a collection of disjoint points. When r is very large, Xr is topologically very similar to X∞, i.e., one giant, featureless blob. As the sequence Xr progresses through the continuum of scales between these two trivial extremes, it undergoes non-trivial topological changes: components merge, and loops form and later get filled. In higher dimensions, cavities of various dimensions likewise form and get filled in.

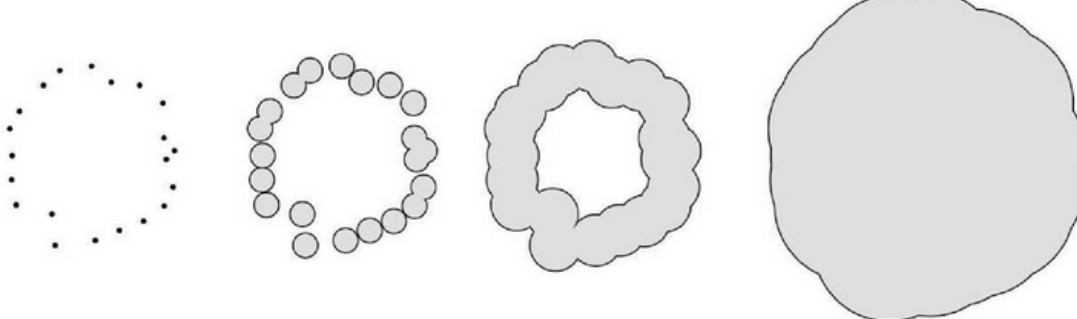

**Scheme 2.** Illustrative 2-dimensional synthetic point cloud.

A small two-dimensional point cloud X viewed at four different scales $0<a<b<c$, forming the filtration $X=X0 \subset Xa \subset Xb \subset Xc$.

In the parlance of TDA, we refer to this appearance and disappearance of topological structures as the birth and death of homology classes in various degrees. We capture the whole life cycle with a mathematical object called the *persistent homology* of the point cloud, which can be fully described by its *persistence diagram*, a planar collection of points (labelled by multiplicity), whose coordinates encode the birth and death of homological features. For the filtration in the figure above, the persistence diagram that tracks one-dimensional features (i.e. unfilled loops) contains only a single point with coordinates (x, y). Here, the first coordinate, x, is the radius at which the loop is first formed, and the second coordinate, y, is the radius at which the loop has just been filled in. In the example it is clear that $a<x<b<y<c$.

As multisets of points in the plane, persistence diagrams are not immediately usable as features for SVMs. One way to vectorise persistence diagrams and thus render them digestible by SVMs is to define kernels based on the diagrams, with the heat kernel (*Reininghaus et al., 2015*) being an oft-used candidate with nice properties. For persistence diagrams P and Q, the heat kernel can informally be defined by the inner product of two solutions of the heat equation—one with an initial condition defined by P, and the other one defined by Q.

In this analysis, we calculated the persistent homology of the alpha complex (*Edelsbrunner and Mücke, 1994*) of the point clouds, using GUDHI (*The GUDHI Editorial Board, 2019*). The heat kernels were computed using RFPKOG (*Spreemann, 2021*). Only the persistence diagrams in degree 1 were used. Since the number of whole CNS point clouds was relatively small, we subsampled each one randomly to 8000 points 100 times, producing a total of 3100 point clouds. This was done both in order to test the stability of the method and to ensure that the variability in the number of points in each cloud is not the source of any signal.

The hyperparameters involved, i.e., the SVM regulariser and the heat kernel bandwidth, were determined by a parameter search in the following way. Six point clouds from males and six from females were randomly selected. All 100 subsampled versions of each of these 12 constituted a training set, for a total of 1200 training point clouds. The remaining 1900 subsampled point clouds constituted the testing set. The Pearson's correlation between the gender predicted by the SVM on

the testing set and the ground truth was computed for each choice of hyperparameters, and a choice in a stable region with high correlation was selected: a regularisation parameter c=10 in the notation of *Pedregosa et al., 2011* and a bandwidth of σ=1/100 in the notation of *Reininghaus et al., 2015*. For the simple distance distribution features, a similar parameter selection process yielded c=10 and a radial kernel bandwidth of 10^5.

## Immunofluorescence and confocal microscopy

Embryos were collected and staged at 25°C on apple agar plates supplemented with yeast paste. Standard methods were used for dechorionation, removal of the vitelline membrane and fixation (*Bashaw, 2010*). Embryos were stored in 100% ethanol at –20°C before IHC labelling. Embryos were stained with mouse anti-myc[9EH10] (1:100, DSHB), visualised with goat anti mouse Alexa Fluor 488 (1:400, Jackson ImmunoResearch) together with conjugated goat anti HRP Alexa Fluor 647 (1:200, Jackson ImmunoResearch) and mounted in VectaShield (Vector Laboratories). Duel colour Z-stack images of stage 15/16 and late stage 17 embryos were obtained on a CSU-W1 Confocal Scanner Unit (Yokogowa, Japan) using two prime BSI express cameras (Teledyne Photometrics). For motor neuron and adult brain preparations, larval fillets from the third instar larvae, or brains from adults were dissected and fixed with 4% formaldehyde (Sigma-Aldrich) for 20 mins. After fixation, samples were rinsed with 1 × PBS and were washed in PBT overnight at 4°C, and then mounted in Vectashield anti-fade mounting medium. Z-stack images were obtained with a Leica SP8 upright confocal microscope.

For Deadpan staining, the CNS from the third instar larvae were dissected out and fixed with 4% PFA for 20 mins. After fixation, samples were rinsed with 1 × PBS, and permealised with PBT (1 × PBS + 4% Triton-X 100). Antibody stainings were done in PBT +5% normal goat serum. The dilution for chicken anti-GFP (Abcam) was 1:500, for rat anti-Deadpan (Abcam) was 1:50. Goat anti-chicken Alexa Fluor 488, and goat anti-rat Alexa Fluor 594 secondary antibodies were obtained from ThermoFisher and used at the 1:500 dilutions. The CNS were mounted in VectaShield antifade mounting medium, and imaged using a Leica SP8 upright confocal microscope.

## Statistical analysis

Column statistics analyses were performed using GraphPad Prism 9 (GraphPad Software). For *Figure 1*, statistical significance was determined by unpaired t-test. For *Figures 2–4*, statistical significances were determined by ordinary one-way ANOVA, followed by a Tukey's honestly significant difference test when multiple comparisons were required. The distribution analysis in *Figure 2* were performed using matlab (MathWorks). Distances between nuclei coordinates were calculated in matlab using code available at https://doi.org/10.5281/zenodo.6574838 and plotted as a histogram of distance distribution.

## Acknowledgements

This work was supported in part using the resources and services of the BioImaging & Optics Platform (BIOP) Research Core Facility at the School of Life Sciences of EPFL and we are especially thankful for the assistance of Arne Seitz. We are grateful to Hugo Bellen, Benjamin White, Gerry Rubin, Vanessa Auld, Christopher Potter, Mel Feany, Serge Birman, Ed Kravitz, Chris Doe, and Pavel Tomancak for generating *Drosophila* stocks, software and providing advice. Stocks obtained from the Bloomington *Drosophila* Stock Center (NIH P40OD018537) were used in this study.

## Additional information

### Funding

| Funder | Grant reference number | Author |
| --- | --- | --- |
| Swiss National Science Foundation | 31003A_179587 | Brian D McCabe |

The funders had no role in study design, data collection and interpretation, or the decision to submit the work for publication.

## Author contributions
Wei Jiao, Conceptualization, Data curation, Formal analysis, Investigation, Methodology, Resources, Validation, Visualization, Writing – original draft, Writing – review and editing; Gard Spreemann, Conceptualization, Investigation, Methodology, Software, Validation, Visualization, Writing – original draft, Writing – review and editing; Evelyne Ruchti, Resources, Visualization; Soumya Banerjee, Resources; Samuel Vernon, Investigation; Ying Shi, Formal analysis, Methodology; R Steven Stowers, Resources, Writing – review and editing; Kathryn Hess, Methodology, Supervision, Writing – original draft, Writing – review and editing; Brian D McCabe, Conceptualization, Funding acquisition, Project administration, Supervision, Writing – original draft, Writing – review and editing

## Author ORCIDs
Wei Jiao  http://orcid.org/0000-0003-1536-5937
Brian D McCabe  http://orcid.org/0000-0003-1620-0501

## Decision letter and Author response
Decision letter https://doi.org/10.7554/eLife.74968.sa1
Author response https://doi.org/10.7554/eLife.74968.sa2

## Additional files

### Supplementary files
- MDAR checklist
- Supplementary file 1. CNS nuclei center of geometry coordinates.

### Data availability
All data generated or analysed during this study are included in the manuscript and supporting files. A source data file has been provided for all figures. Raw co-ordinates of the center of geometry of CNS nuclei for all animals is attached as Supplementary file 1. Matlab code employed is available at https://doi.org/10.5281/zenodo.6574838. Example unprocessed whole CNS microscopy data is publicly available for neurons - https://doi.org/10.5281/zenodo.5585334 and for glia https://doi.org/10.5281/zenodo.5585358.

The following datasets were generated:

| Author(s) | Year | Dataset title | Dataset URL | Database and Identifier |
|---|---|---|---|---|
| McCabe B, Jiao W | 2021 | Whole Brain Drosophila Larval Neurons | https://doi.org/10.5281/zenodo.5585334 | Zenodo, 10.5281/zenodo.5585334 |
| McCabe B, Jiao W | 2021 | Whole Brain Drosophila Larval Glia | https://doi.org/10.5281/zenodo.5585358 | Zenodo, 10.5281/zenodo.5585358 |

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

# Appendix 1

## Key resources table

### Appendix 1—key resources table

| Reagent type (species) or resource | Designation | Source or reference | Identifiers | Additional information |
|---|---|---|---|---|
| strain, strain background (*Escherichia Coli*) | One shot top10 | Invitrogen | Cat#: C404010 | |
| genetic reagent (*D. melanogaster*) | y[1] w[*]; Mi{y[+mDint2]= MIC}Syt1[MI02197] | Bloomington *Drosophila* Stock CenterPMID: 21985007 | BDSC:35973FLYB: FBal0314405 RRID:BDSC_35973 | FlyBase symbol: Mi{y[+mDint2]= MIC}Syt1[MI02197] |
| genetic reagent (*D. melanogaster*) | y[1] w[*] Mi{y[+mDint2]= MIC}Sh[MI10885] | Bloomington *Drosophila* Stock CenterPMID: 21985007 | BDSC:56260FLYB: FBal0297530 RRID:BDSC_56260 | FlyBase symbol: Mi{MIC} ShMI10885 |
| genetic reagent (*D. melanogaster*) | y[1] w[*];Mi{y[+mDint2]= MIC}Shal[MI10881] | Bloomington *Drosophila* Stock CenterPMID: 21985007 | BDSC:56089FLYB: FBal0295200 RRID:BDSC_56089 | FlyBase symbol: Mi{MIC} ShalMI10881 |
| genetic reagent (*D. melanogaster*) | y[1] w[*]; Mi{y[+mDint2]=MIC} Shab[MI00848] | Bloomington *Drosophila* Stock CenterPMID: 21985007 | BDSC:34115FLYB: FBal0249123 RRID:BDSC_34115 | FlyBase symbol: Mi{MIC}ShabMI00848 |
| genetic reagent (*D. melanogaster*) | nSyb-GAL4 | Bloomington *Drosophila* Stock CenterPMID: 18621688 | BDSC: 39171FBgn0013342 RRID:BDSC_39171 | Flybase symbol: P{GMR57C10-GAL4} |
| genetic reagent (*D. melanogaster*) | repo-GAL4 | Bloomington *Drosophila* Stock Center PMID: 7926782 | BDSC:7415FLYB: FBal0127275 RRID:BDSC_7415 | FlyBase symbol: P{GAL4}repo |
| genetic reagent (*D. melanogaster*) | repo-QF2 | Bloomington *Drosophila* Stock Centerdoi:10.1534/ genetics.116.191783 | BDSC:66477FLYB: FBal0322908 RRID:BDSC_66477 | FlyBase symbol: P{ET-QF2. GU}repo |
| genetic reagent (*D. melanogaster*) | Shaw-GAL4 | Bloomington *Drosophila* Stock CenterPMID: 21985007 | BDSC:60325FLYB: FBal0304243 RRID:BDSC_60325 | FlyBase symbol: GAL4Shaw- MI01735-TG4.1 |
| genetic reagent (*D. melanogaster*) | Ddc-GAL4 | Bloomington *Drosophila* Stock Center doi:10.1006/ dbio.1994.1261 | BDSC:7009FLYB: FBtp0012451 RRID:BDSC_7009 | FlyBase symbol: P{Ddc-GAL4.L} |
| genetic reagent (*D. melanogaster*) | TH-GAL4 | Bloomington *Drosophila* Stock CenterPMID: 12555273 | BDSC:8848FLYB: FBtp0114847 RRID:BDSC_8848 | FlyBase symbol: P{ple-GAL4.F}3 |
| genetic reagent (*D. melanogaster*) | Trh-GAL4 | Bloomington *Drosophila* Stock Centerdoi:10.1371/journal. pone.0010806 | BDSC:38389FLYB: FBtp0055412 RRID:BDSC_38389 | FlyBase symbol: P{Trh-GAL4.long} |
| genetic reagent (*D. melanogaster*) | UAS_H2A-GFP | Steve Stowers (Montana SU) doi:10.1371/journal. pgen.1008609 | FLYB: FBgn0001196 | |
| genetic reagent (*D. melanogaster*) | QUAS_H2A-mCherry | Steve Stowers(Montana SU) doi:10.1371/journal. pgen.1008609 | FLYB: FBgn0001196 | |
| genetic reagent (*D. melanogaster*) | brp-GAL4 | This paper, available upon request, https://www. epfl.ch/ labs/mccabelab/resources/ | FLYB:FBgn0259246 | See Materials and Methods, Section 2 |
| genetic reagent (*D. melanogaster*) | Syt1-Gal4 | This paper, available upon request, https://www. epfl.ch/ labs/mccabelab/resources/ | FLYB: FBal0314405 | See Materials and Methods, Section 4 |
| genetic reagent (*D. melanogaster*) | Sh-Gal4 | This paper, available upon request,https://www. epfl.ch/ labs/mccabelab/resources/ | FLYB: FBal0297530 | See Materials and Methods, Section 4 |
| genetic reagent (*D. melanogaster*) | Shal-Gal4 | This paper, available upon request,https://www.epfl.ch/ labs/mccabelab/resources/ | FLYB: FBal0249123 | See Materials and Methods, Section 4 |
| genetic reagent (*D. melanogaster*) | Shab-Gal4 | This paper, available upon request,https://www. epfl.ch/ labs/mccabelab/resources/ | FLYB: FBal0249123 | See Materials and Methods, Section 4 |

*Appendix 1 Continued on next page*

*Appendix 1 Continued*

| Reagent type (species) or resource | Designation | Source or reference | Identifiers | Additional information |
|---|---|---|---|---|
| genetic reagent (*D. melanogaster*) | UAS_H2A::GFP-T2A-mKok::Caax | This paper, available upon request, https://www.epfl.ch/labs/mccabelab/resources/ | FLYB: FBgn0001196 | See Materials and Methods, Section 3 |
| antibody | anti-myc9EH10(Mouse monoclonal) | DSHB | DSHB Cat# 9E 10, RRID:AB_2266850 | IF(1:100) |
| antibody | anti-GFP(Chicken polyclonal) | Abcam | Cat#ab13970 | IF(1:500) |
| antibody | anti-Deadpan(Rat monoclonal) | Abcam | Cat#ab195173 | IF(1:50) |
| antibody | Goat Anti-Mouse IgG (H+L), Alexa Fluor 488 (Goat polyclonal Secondary Antibody) | Jackson ImmunoResearch | Cat#115-545-166 | IF(1:400) |
| antibody | Goat Anti-Horseradish Peroxidase,Alexa Fluor 647 (Goat polyclonal) | Jackson ImmunoResearch | Cat#123-605-021 | IF(1:200) |
| antibody | Goat anti-Chicken IgY (H+L), Alexa Fluor 488 (Goat polyclonal Secondary Antibody) | ThermoFisher | Cat#A-11039 | IF(1:500) |
| antibody | Goat anti-Rat IgG (H+L) Cross-Adsorbed, Alexa Fluor 594 (Goat polyclonal Secondary Antibody) | ThermoFisher | Cat#A-11007 | IF(1:500) |
| recombinant DNA reagent | pBID_DSCP-G-Gal4 (plasmid) | McCabe Lab,Available upon request.https://www.epfl.ch/labs/mccabelab/resources/ | Cat# #35,200 | |
| recombinant DNA reagent | pJFRC81-10XUAS-IVS-Syn21-GFP-p10 (plasmid) | Addgene | Cat#: 36,432 | |
| recombinant DNA reagent | pHD-sfGFP Scareless dsRed (plasmid) | Addgene | Cat#: 80,811 | |
| recombinant DNA reagent | pCFD3 (plasmid) | Addgene | Cat#: 49,410 | |
| recombinant DNA reagent | pCR4 brp-Gal4 | This paper, available upon request, https://www.epfl.ch/labs/mccabelab/resources/ | | CRISPR construct inserted in *D. Melanogaster* |
| recombinant DNA reagent | pBID LexO_H2A-mCherry | Gift from Steve Stowers (Montana SU), DOI: 10.1371/journal.pgen.1008609 | | |
| recombinant DNA reagent | pCS2+ChMermaid S188 | Addgene | Cat#: 53,617 | |
| recombinant DNA reagent | pBID-UAS_H2A::GFP-T2A-mKok::Caax | This paper, available upon request, https://www.epfl.ch/labs/mccabelab/resources/ | | construct inserted in *D. Melanogaster* |
| sequence-based reagent | pCFD3 gRNA brp-5' | This paper | Guide RNA for Brp CRISPR knock in | GGTGAACCGACCGGGACAAC |
| sequence-based reagent | pCFD3 gRNA brp-3' | This paper | Guide RNA for Brp CRISPR knock in | GGGAGCCCCGCGACCGCTCC |
| sequence-based reagent | brp Ha1 Fo | This paper | PCR primer | GAGAGAGCATCTCGATTGTGCCGTGTG |
| sequence-based reagent | brp Pam7 Re | This paper | PCR primer | AATGTTGTCCCGGTCGGTTCACCG |

*Appendix 1 Continued on next page*

*Appendix 1 Continued*

| Reagent type (species) or resource | Designation | Source or reference | Identifiers | Additional information |
|---|---|---|---|---|
| sequence-based reagent | brp Pam7_In1 Re | This paper | PCR primer | TTCTAGCGTCCAA CGGCTCAGCTGTG GGCCATTTTCTAGT AATGTTGTCCCGG TCGGTTCACCG |
| sequence-based reagent | brp HA_In1 Fo | This paper | PCR primer | ACTAGAAAATGGCC CACAGCTGAGCC |
| sequence-based reagent | brp V5_In1 Re | This paper | PCR primer | TAGAATCGAGACCG AGGAGAGGGTTAGGG ATAGGCTTACCCATT GCTGAAATTCACACA CACACAGAATTCATGAG |
| sequence-based reagent | brp V5_ Fo | This paper | PCR primer | GGTAAGCCTATCCC TAACCCTCTCCTC |
| sequence-based reagent | brp PB5' Re | This paper | PCR primer | TTAAGGGATCTTTCTA TTAGTATAACACTGCATGC |
| sequence-based reagent | brp Ex2 fo | This paper | PCR primer | AAATTGCATGCAGTGTT ATACTAATAGAAAGATCC CTTAATCGGCAGTCCAT ACTACCGCGACATGGATG |
| sequence-based reagent | brp Pam2_Re | This paper | PCR primer | TCTGGAGCGGT CGCGGGGC |
| sequence-based reagent | brp Pam2_Brp Ha2 Re | This paper | PCR primer | GCTCGTCCTCTAGGTAC AGGCCCCGTTCGAGGGA TCTGTCTCTGGAGC GGTCGCGGGG |
| sequence-based reagent | brp Ha2 Fo | This paper | PCR primer | GACAGATCCCTC GAACGGGGCC |
| sequence-based reagent | Syn21 H2A Fo | This paper | PCR primer | AACTTAAAAAAAAAA ATCAAAATGTCTGGA CGTGGAAAAGGTGGC |
| sequence-based reagent | H2A Re | This paper | PCR primer | CCCAAGAAGACC GAGAAGAAGGCC |
| sequence-based reagent | H2A-GFP Fo | This paper | PCR primer | ACAGGCTGTTCTGT TGCCCAAGAAGACC GAGAAGAAGGCCAT GGTGTCCAAGGG CGAGGAG |
| sequence-based reagent | GFP-T2A Re | This paper | PCR primer | GGGTTCTCCTCCAC ATCGCCGCAGGTCAG CAGGCTGCCGCGGC CCTCCTTGTACAGCT CATCCATGCCCAGG |
| sequence-based reagent | T2A-mKok Fo | This paper | PCR primer | GCGGCAGCCTGCT GACCTGCGGCGATG TGGAGGAGAACCCC GGGCCCATGGTGAGT GTGATTAAACCAG AGATGAAGATG |
| sequence-based reagent | mKok-Caax Re | This paper | PCR primer | TTACATAATTACACA CTTTGTCTTTGACTT CTTTTTCTTCTTTTTA CCATCTTTGCTCATGG AATGAGCTACTGCAT CTTCTACCTGC |
| chemical compound, drug | Formaldehyde 37% | Sigma | Cat#: 252,549 | |
| chemical compound, drug | Low melt agarose | Peq gold | Cat#: 35–2010 | |
| chemical compound, drug | VECTASHIELD Antifade Mounting Media | VECTOR Laboratories | Cat#: H-1000 | |
| chemical compound, drug | FluoSpheres, 0.2 μm, red fluorescent (580/605) | Thermo Fisher Scientific | Cat#: F8810 | |

*Appendix 1 Continued on next page*

*Appendix 1 Continued*

| Reagent type (species) or resource | Designation | Source or reference | Identifiers | Additional information |
|---|---|---|---|---|
| commercial assay or kit | Zero Blunt TOPO PCR Cloning Kit | Invitrogen | Cat#: 450,245 | |
| commercial assay or kit | pCR8/GW/TOPO TA Cloning Kit | Invitrogen | Cat#: K250020 | |
| commercial assay or kit | Gateway LR Clonase II Enzyme mix | Invitrogen | Cat#:11791020 | |
| commercial assay or kit | NEBuilder HiFi DNA Assembly Master Mix | New England Biolabs | Cat#: E2621S | |
| commercial assay or kit | KLD enzyme mix | New England Biolabs | Cat#:M0554S | |
| commercial assay or kit | Platinium Superfi polymerase | Invitrogen | Cat#:12359010 | |
| software, algorithm | CNS nuclei distance (code for Matlab) | https://doi.org/10.5281/zenodo | | This manuscript |
| software, algorithm | Fiji | https://fiji.sc/ | RRID:SCR_002285 | |
| software, algorithm | Arivis Vision4D 3.0.0 | Arivis | RRID:SCR_018000 | |
| software, algorithm | MATLAB (R2018a) | MathWorks | RRID:SCR_001622 | |
| software, algorithm | GraphPad Prism 9.0 | GraphPad | RRID:SCR_002798 | |

