## [Editor Report]

This manuscript describes a pipeline involving whole brain imaging, automated neuronal segmentation, and topographical analysis, to assess the number of specific cell types in the larval *Drosophila* brain. The authors uncover unexpected sexual dimorphism at this early stage. This paper will be of interest to neuroscientists – from those who use larval *Drosophila* as their study model to others who are generally interested in connectomics and transcriptomics.

---

## [Decision Letter]

**Decision letter after peer review:**

Thank you for submitting your article "Intact *Drosophila* Whole Brain Cellular Quantitation reveals Sexual Dimorphism" for consideration by *eLife*. Your article has been reviewed by 2 peer reviewers, including Sonia Sen as Reviewing Editor and Reviewer #1, and the evaluation has been overseen by K VijayRaghavan as the Senior Editor.

Essential revisions:

While noting the value and general rigour of this work, we had a few concerns that the authors could address. These are listed below.

1) The new Gal4 lines: These lines form the basis of this study and we anticipate that they will be of value to the broader community. For both these reasons, we request that the authors verify them better, for example, with neuroblast-specific markers and/or co-labels with markers of mature neurons.

2) Comparison with previous studies: We were concerned about the comparison with previous studies – please see detailed reviews below. Could the authors please revisit the text surrounding this, and perhaps provide a systematic comparison of the previous studies that have 'counted' cell numbers?

3) Interpretation of some of the findings: Some of the findings are interesting, but we were not sure what they meant (please see the detailed comments below). Could the authors discuss this? If they feel it would be too speculative, they could perhaps use the Ideas and Speculation section within discussion to do this.

4) Description of methods: Could the authors please describe some of their methods in more detail? Please see the detailed reviews below, but, for example, the procedures leading to the topographical comparisons, or the comparison of 3D coordinates across brains.

*Reviewer #1 (Recommendations for the authors):*

– The authors have generated invaluable Gal4 lines through their study. We would have liked to have seen a better characterization of them particularly because we think they will be of value to the broader community. For example with neuroblast-specific markers or co-labels with markers of mature neurons.

– It would be nice to have a more systematic description of previous studies that have estimated cell numbers in larval and adult brains. The authors have done this, but the larval and adult studies are listed together, making it hard to assess the different approaches and their advantages and limitations. Could the authors please look into discussing it systematically in this way?

– The authors quote previous studies suggesting glia constitute 10% of the cell in the brain – this is for the adult, isn't it? Significant neurogenesis occurs between early L3 and the adult and so is it possible that the 30% observed in this study transitions to 10%?

– I did not quite understand how one can compare 3D coordinates of nuclei across brains if there isn't a common reference? Could the authors please elaborate on this?

– What do the authors interpret from the nuclear distance distribution patterns of neurons and glia?

– Similarly, the authors say that the sexual dimorphism extends to deeper features, is subtle, but widespread. Could the authors speculate what these subtle, widespread features might be?

*Reviewer #2 (Recommendations for the authors):*

The conclusions of this paper are supported mainly by data, but some aspects of data analysis and interpretation need to be clarified and extended to solidify their findings.

1) The paper is generally written well, but some data interpretations are based on inaccurate and shaky estimates, which are better avoided. In my understanding, none of the precedents formally publish the number of neurons and glia in the whole nervous system of third instar larvae. As mentioned in the text, there are indeed some estimates that range from 10,000 to 15,000 (P3, L15, Scott, et al., 2001), but they are not published as data and are not of the same age as their animal. Therefore, it is better for the authors to avoid direct comparisons with this wide range of statistics to draw numerical arguments (e.g., P8. L137-138). Instead, those studies collectively suggest the present results are in good agreement with them, solidifying their work rather than contradicting.

2) Usage of the term "brain" has to be strict to avoid any future confusion (cf. title and abstract). While I agree that they found sexual dimorphism in the larval nervous system, I am unsure whether it is about the brain. The central nervous system of larvae consists of the brain and ventral nervous system, and the methodology of the current study analyzes both. Numerically speaking, the latter is the majority (4~5 times more), suggesting that the difference is somewhat related to other peripheral parts of the nervous system than the brain. I suggest the authors revise the relevant points and put their work in the proper context.

3) Text writing in the methodological part could work better to substantiate their main findings further. In particular, the procedures leading to the topographical comparisons that contribute to their main conclusion (P8. 4th paragraph – P9. 2nd paragraph) uses many unfamiliar mathematical terms without proper explanations that make it difficult for non-specialists to follow.

4) In the distribution of inter-nuclei distances (Figure 2e), I see fractions of the cells fall into the range of distance 0, suggestive of double counting. Some questions about nuclei counting accuracy.

5) I would like to know the specific citation behind their notion that "genes likely to be expressed only in active larval neurons with synaptic connections but not by neuroblasts or by immature neurons". Brp is known to be expressed in the developing embryo (Wagh et al., Neuron, 2006).

6) (P9. L192) "As the Sh gene is x-linked, we carried out our quantification analysis in male brains only to avoid potential gene dosage effects."

How does the dosage effect affect their cell counting? In other words, is this data appropriate being here without contributing the main conclusion of sexual dimorphism?

7) A list of questions for methodological clarifications.

– What genotypes were used for the topographical comparisons (Figure 3f 3g)?

– (P8. L160) What are the "multi-scale topological structures"?

– (P8. L162-164) I do not follow this part. The authors need to elaborate on the explanation.

– What is the voxel resolution of their images?

– How are inter-nuclei distances calculated (Figure 2e)?

– (P16. L356) How is the volume cutoff (15um3) determined?

– What is coded by the color in Figure 2a-2d and others?

– What is the unit of the abscissa in Figure 2e?

– (P8. L137) What is the denominator to get the value (~15 to 30%)?

8) (Figure 1, 2, 3, 4) GeneName>GAL4. Is this the standard way to describe the genotype?

---

## [Author Response]

Reviewer #1 (Recommendations for the authors):– The authors have generated invaluable Gal4 lines through their study. We would have liked to have seen a better characterization of them particularly because we think they will be of value to the broader community. For example with neuroblast-specific markers or co-labels with markers of mature neurons.

We thank the reviewer for their kind and insightful comments. To address their concern, we have carried out several additional experiments to further characterise these lines. We first examined the expression of these lines during embryo development (new Figure 2—figure supplement 2). We observed no expression at stage 15-16 or stages prior to this (Figure 2—figure supplement 2a). However, at stage 17 all three lines had neuronal expression (Figure 2—figure supplement 2a). This is in good agreement with the onset of synaptic activity in embyros (Baines and Bate 1998). We also examined the expression of Deadpan, a transcription factor that labels neuroblast nuclei, in the larval CNS together with histoneGFP expression labelling driven by all three neuronal lines. Consistent with our lines not labelling neuroblasts, we found that the expression of *brp*-Gal4, *Syt1*-Gal4 and *nSyb*-Gal4 were mutually exclusive from Deadpan labelling (Figure 2—figure supplement 2b). Finally, we also examined the expression of all three lines in the adult CNS and found that all three also appear to express in neurons in the adult brain (Figure 2—figure supplement 2c).

– It would be nice to have a more systematic description of previous studies that have estimated cell numbers in larval and adult brains. The authors have done this, but the larval and adult studies are listed together, making it hard to assess the different approaches and their advantages and limitations. Could the authors please look into discussing it systematically in this way?

As correctly described by reviewer #2, most of these prior estimates seem to be rough approximations and are not based on data or at least not explicitly linked to data in the source manuscripts. We have however rewritten this section as the reviewer requested. Larval and adult estimates are now separated and additional references added to support the range of numbers proposed (lines 16-23 revised manuscript).

– The authors quote previous studies suggesting glia constitute 10% of the cell in the brain – this is for the adult, isn't it? Significant neurogenesis occurs between early L3 and the adult and so is it possible that the 30% observed in this study transitions to 10%?

This is an excellent point. We have added this potential explanation of divergence in our measurements to this discussion (line 305-309 revised manuscript).

– I did not quite understand how one can compare 3D coordinates of nuclei across brains if there isn't a common reference? Could the authors please elaborate on this?

Reviewer #2 also asked for clarification about this novel biological data analysis method. Topological summaries indeed lack common reference points but nonetheless can be compared by standard methods. They can be thought of as multi-scale descriptions of the shape of the data set. In addition to adding several basic references to topological data analysis to aid biologists, we added this ‘motivating example’ or thought experiment to help illustrate the underlying principle to the mathematics methods section (revised manuscript lines 401-409)

“For a motivating example of the principles underlying topological summaries, one could think of pearls forming a necklace. Topological summaries express the global structure of the necklace formed by the relationships between the positions of the individual pearls, but are invariant under translation and rotation of the necklace overall. Two such necklaces have topological summaries that are comparable even if the pearls in one have no relationship to the pearls in the other. It is the global structures – such as its circular shape on a large scale, or bulges on a smaller scale – formed by the relationships of the individual pearls of each one that matter”

– What do the authors interpret from the nuclear distance distribution patterns of neurons and glia?

We interpret these results as yet another confirmation that cells labelled are different between the neuronal Gal4 lines and the glial lines. We have edited the manuscript to make this point explicit (line 135-137 revised manuscript)

– Similarly, the authors say that the sexual dimorphism extends to deeper features, is subtle, but widespread. Could the authors speculate what these subtle, widespread features might be?

We are very reluctant to speculate as we have no data on which to propose a hypothesis. Topology analysis is telling us the differences between the female and male CNS is subtle, widespread and clearly detectible but also difficult to identify with simpler analysis approaches. An interesting future direction might be to examine CNS gynandromorphs to determine if these differences could be more discreetly localised.

Reviewer #2 (Recommendations for the authors):The conclusions of this paper are supported mainly by data, but some aspects of data analysis and interpretation need to be clarified and extended to solidify their findings.

We thank the reviewer for their supportive comments and have sought to address their concerns.

1) The paper is generally written well, but some data interpretations are based on inaccurate and shaky estimates, which are better avoided. In my understanding, none of the precedents formally publish the number of neurons and glia in the whole nervous system of third instar larvae. As mentioned in the text, there are indeed some estimates that range from 10,000 to 15,000 (P3, L15, Scott, et al., 2001), but they are not published as data and are not of the same age as their animal. Therefore, it is better for the authors to avoid direct comparisons with this wide range of statistics to draw numerical arguments (e.g., P8. L137-138). Instead, those studies collectively suggest the present results are in good agreement with them, solidifying their work rather than contradicting.

This point was also raised Reviewer #1. The reviewer is correct that previous estimates were not published as data and have a broad range. Nonetheless they have been widely cited with the most recent estimates in higher range ~15K (e.g. Meinertzhagen, 2018; Eschbach and Zlatic, 2020).

We have rewritten this section (lines 13-23 revised manuscript) to make the point that the cited numbers are approximations more explicit and revised our conclusions as the reviewer suggested to indicate our data supports neuron numbers at the lowest end of previous estimates. As the reviewer suggested, we have removed all direct numerical comparisons to previous approximations.

2) Usage of the term "brain" has to be strict to avoid any future confusion (cf. title and abstract). While I agree that they found sexual dimorphism in the larval nervous system, I am unsure whether it is about the brain. The central nervous system of larvae consists of the brain and ventral nervous system, and the methodology of the current study analyzes both. Numerically speaking, the latter is the majority (4~5 times more), suggesting that the difference is somewhat related to other peripheral parts of the nervous system than the brain. I suggest the authors revise the relevant points and put their work in the proper context.

The reviewer is absolutely correct. We have edited the title, abstract, text and figure legends to use the term ‘central nervous system (CNS)’ when appropriate, which is the case for the majority of our experiments. We make explicit that the *Drosophila* CNS and our experiments include measurements of the brain and VNC combined (line 13-16 revised manuscript).

3) Text writing in the methodological part could work better to substantiate their main findings further. In particular, the procedures leading to the topographical comparisons that contribute to their main conclusion (P8. 4th paragraph – P9. 2nd paragraph) uses many unfamiliar mathematical terms without proper explanations that make it difficult for non-specialists to follow.

Indeed, there are many terms here that may be unfamiliar to those of us outside mathematics fields. To address the reviewers concern, we have made a number of edits. In the main text, we have added additional accessible references to serve as general primers for non-specialists and added an additional explanation of the general approach (lines 177-183 revised manuscript). We have also expanded the methods section to include further general topology references and a ‘motivating example’ or thought experiment to help illustrate the underlying principle given that this type of topological data analysis is new for neuroscience microscopy data (lines 401-409 revised manuscript).

4) In the distribution of inter-nuclei distances (Figure 2e), I see fractions of the cells fall into the range of distance 0, suggestive of double counting. Some questions about nuclei counting accuracy.

The plot in Figure 2e is derived from a regression analysis (the raw inter-nuclei distance values are provided in supplementary data1 and the regression analysis matlab code is publicly available at https://doi.org/10.5281/zenodo.6574838). As the reviewer correctly points out the input data are distances, so we should have should have set the range of the analysis outputs to positive values. We have rerun the analysis making this correction and edited the text and figure.

5) I would like to know the specific citation behind their notion that "genes likely to be expressed only in active larval neurons with synaptic connections but not by neuroblasts or by immature neurons". Brp is known to be expressed in the developing embryo (Wagh et al., Neuron, 2006).

This point was also raised Reviewer #1. To address this concern, we have carried out several additional experiments to further characterise these lines. We first examined the expression of these lines during embryo development (new Figure 2—figure supplement 2). We observed no expression at stage 15-16 or stages prior to this (Figure 2—figure supplement 2a). However, at stage 17 all three lines had expression (Figure 2—figure supplement 2a). This is in good agreement with the onset of synaptic activity in embyros (Baines and Bate 1998).

We have edited the statement cited to read ‘genes likely to be expressed only in active neurons with synaptic connections but not by neuroblasts or by immature neurons’ (line 90 revised manuscript) to define our rationale and subsequently confirm these lines indeed initiate expression when synaptic activity begins (new Figure 2—figure supplement 2) and lines 110-121 in the revised manuscript.

6) (P9. L192) "As the Sh gene is x-linked, we carried out our quantification analysis in male brains only to avoid potential gene dosage effects."How does the dosage effect affect their cell counting? In other words, is this data appropriate being here without contributing the main conclusion of sexual dimorphism?

Our goal with this series of experiments was to illustrate the utility and application of the technique to investigate the frequency of expression of a gene family throughout the CNS and we believe we have made some surprising observations into the relative abundance of K^+^ channels. Our concern was that well documented X-linked gene dosage effects could influence expression of the *Sh*-Gal4 line (which in on the X chromosome) which is why we choose to carry out these experiments in only one sex. We think these experiments are useful to illustrate another utility of the technique independently of other findings related to sexual dimorphism in the manuscript. We have edited the abstract to make this additional goal clear.

7) A list of questions for methodological clarifications.– What genotypes were used for the topographical comparisons (Figure 3f 3g)?

All point clouds from all CNS lines were used. This has been clarified to the methods (revised manuscript line 413)

– (P8. L160) What are the "multi-scale topological structures"?

We have added additional references that explain these terms, see also answer to question 3 above.

– (P8. L162-164) I do not follow this part. The authors need to elaborate on the explanation.

We have added additional explanation in the main text and methods, see also answer to question 3 above.

– What is the voxel resolution of their images?

Our voxel resolution is 0.317um. We had added this to the methods (revised manuscript line 383)

– How are inter-nuclei distances calculated (Figure 2e)?

Arivis4D outputs the three dimensional coordinates of the geometric center of every labelled nuclei. These co-ordinates were then processed using a Matlab code we generated to calculate internuclei the distance in 3D volume. This code is publicly available to download at https://doi.org/10.5281/zenodo.6574838 and this has been added to the revised methods.

– (P16. L356) How is the volume cutoff (15um3) determined?

The volume was calculated by as the component of the radius of a the nucleus -1.5um producing a spherical volume – 15um3 (V=4-3×*π*r^!^). This was empirically validated by volume measurements of subsets of neurons. This has been clarified in the methods.

– What is coded by the color in Figure 2a-2d and others?

Colour codes Z-coordinate position in these 2D images. We have explained this in the revised figure legends and added the colour scale to the methods (line 399).

– What is the unit of the abscissa in Figure 2e?

um. We have corrected this error.

– (P8. L137) What is the denominator to get the value (~15 to 30%)?

We have removed this statement and calculation as described above based on the reviewers valid criticism.

8) (Figure 1, 2, 3, 4) GeneName>GAL4. Is this the standard way to describe the genotype?

Unfortunately, there is no ‘standard’ way to describe these shorthand genotypes (full genotypes are included in the methods and key resources file). A **-**,. , **_** or > have been used to describe a promoter upstream of a Gal4 in various manuscripts. However, as suggested, we have replaced > with – as this seems the most widely employed shorthand.

References

Baines RA, Bate M. 1998. Electrophysiological Development of Central Neurons in the *Drosophila* Embryo. *J Neurosci* 18:4673–4683. doi:10.1523/JNEUROSCI.18-12-04673.1998